**Increasing soil carbon stocks in eight permanent forest plots in China**
**Jianxiao Zhu[1,2], Chuankuan Wang[3], Zhang Zhou[2,4], Guoyi Zhou[5], Xueyang Hu[2], Lai**
**Jiang[2], Yide Li[4], Guohua Liu[6], Chengjun Ji[2], Shuqing Zhao[2], Peng Li[2], Jiangling Zhu[2],**
**Zhiyao Tang[2], Chengyang Zheng[2], Richard A. Birdsey[7], Yude Pan[8], and Jingyun Fang[2]**
[1]State Key Laboratory of Grassland Agro-ecosystems, College of Pastoral Agricultural
Science and Technology, Lanzhou University, Lanzhou 730020, China
[2]Department of Ecology, College of Urban and Environmental Science, and Key Laboratory
for Earth Surface Processes of the Ministry of Education, Peking University, Beijing 100871,
China
[3]Center for Ecological Research, Northeast Forestry University, 26 Hexing Road, Harbin
150040, China
[4]Research Institute of Tropical Forestry, Chinese Academy of Forestry, No. 682 Guangshanyi
Road, Tianhe District, Guangzhou 510520, China
[5]Key Laboratory of Vegetation Restoration and Management of Degraded Ecosystems, South
China Botanical Garden, Chinese Academy of Sciences, Guangzhou 510650, China
[6]Research Center for Eco-Environmental Sciences, Chinese Academy of Sciences, Beijing
100085, China
[7]Woods Hole Research Center, Falmouth, MA 02540, USA
[8]US Department of Agriculture Forest Service, Durham, NH 03824, USA
**Correspondence:** Jingyun Fang (jyfang@urban.pku.edu.cn)
**Abstract.** Forest soils represent a major stock of organic carbon (C) in the terrestrial
biosphere, but the dynamics of soil organic C (SOC) stock are poorly quantified, largely due
to lack of direct field measurements. In this study, we investigated the 20-year changes in
SOC stocks in eight permanent forest plots, which represent boreal (1998–2014), temperate
(1992–2012), subtropical (1987–2008), and tropical forest biomes (1992–2012) across China.
SOC contents increased significantly from the 1990s to the 2010s, mostly in the upper 0–20
cm soil depth, and soil bulk densities do not change significantly during the same period. As a
result, the averaged SOC stocks increased significantly from $125.2 \pm 85.2$ Mg C ha$^{-1}$ in the
1990s to $133.6 \pm 83.1$ Mg C ha$^{-1}$ in the 2010s across the forest plots, with a mean increase of
127.2–907.5 kg C ha$^{-1}$ yr$^{-1}$. This SOC accumulation resulted primarily from increasing leaf
litter and fallen logs, which accounts 3.6–16.3% of above-ground net primary production. Our
findings provided direct evidence that China's forest soils have been acting as significant C
sinks, although their strength varies in forests with different climates.

## 1 Introduction

Terrestrial ecosystems have absorbed approximately 30% of the carbon dioxide ($CO_2$) emitted from human activities since the beginning of the industrial era (IPCC, 2013). Forests have contributed more than half of these carbon (C) fluxes of terrestrial ecosystems (Pan et al., 2011). Since soils contain a huge C stock in forest ecosystems, even a slight change in this stock will induce a considerable feedback to atmospheric $CO_2$ concentrations (Lal, 2004; Luo et al., 2011). Thus, accurate assessment of the changes in soil organic carbon (SOC) is critical to understanding how forest soils will respond to global climate change. However, it is difficult to capture the SOC change with short-term measurements (Smith, 2004) because the soil C pool typically has a longer turnover time and higher spatial variability compared to the vegetation C pool (Schrumpf et al., 2011; Canadell and Schulze, 2014).

Previous efforts have estimated the changes in regional SOC stocks with indirect approaches, such as regional assessments (Yang et al., 2014) and model simulations (Todd-Brown et al., 2013). These estimates often involve large uncertainties due to the inherently high spatial variability of soils and lack of direct measurements representing large areas (Sitch et al., 2013). One reliable approach to reducing the uncertainties is to conduct long-term monitoring of forest SOC stocks at sites that represent broader landscapes (Prietzel et al., 2016). Unfortunately, such repeated, accurate field-based measurements of SOC stocks from which to generate change estimates are generally lacking and inadequate worldwide (Zhao et al., 2019).

A few soil resampling studies have explored SOC changes in different forests, but the results are often contradictory. For instance, Schrumpf et al. (2014) found that SOC in deciduous broadleaved forests in central Germany increased, with a change rate of 650.0 kg C ha$^{-1}$ yr$^{-1}$ from 2004 to 2009. In contrast, Prietzel et al. (2016) indicated that SOC stocks in German forests decreased significantly, with average change rates of 988.2 kg C ha$^{-1}$ yr$^{-1}$ in

forests in the Alps between 1986 and 2011, and 441.1 kg C ha$^{-1}$ yr$^{-1}$ in the Berchtesgaden
region between 1976 and 2011. Kiser et al. (2009) found that the hardwood forest soils in
central Tennessee, USA, exhibited a slight C source, and that the relative change rate ranged
from -0.4% yr$^{-1}$ to 0.3% yr$^{-1}$ between 1976 and 2006. Chen et al. (2015) synthesized global
SOC changes, and found that the relative rates of change in forest SOC stocks were
contradictory among long-term experiments (0.2% yr$^{-1}$), regional comparisons (0.3% yr$^{-1}$),
and repeated soil samplings (-0.1% yr$^{-1}$). Such discrepancies can be partly attributed to
insufficient observations and inconsistent methodologies. The different effects of changing
environmental factors and nitrogen inputs on soil C dynamics may also be involved (Norby
and Zak, 2011). In addition, to date these studies have primarily been conducted in the forests
of Europe and the USA, but few have been carried out in China's forests.

Forests in China cover an area of 156 Mha (Guo et al., 2013), and range from boreal

coniferous forests and deciduous broadleaved forests in the northeast to tropical rain forests
and evergreen broadleaved forests in the south and southwest. They include almost all major
forest biomes of the Northern Hemisphere (Fang et al., 2012). Such variations in climate and
forest types have provided ideal opportunities to examine the spatial patterns of SOC in
relation to meteorological and biological factors. At the national scale, the mean annual air
temperature of China increased by more than 1 ℃ between 1982 and 2011, which is
considerably higher than the global average (Fang et al., 2018). Since the 1980s, the
Government of China has implemented several large-scale national forest protection projects.
These climatic changes and conservation practices in China have significantly stimulated C
uptake into forest ecosystems (Fang et al., 2014, 2018; Feng et al., 2019). Several studies
have assessed the temporal dynamics of SOC stock across China's forests, using model
simulations (Piao et al., 2009) or regional assessments (Pan et al., 2011; Tang et al., 2018).
However, these estimates revealed contrasting trends in SOC dynamics and also lacked direct
measurements of SOC change.
Therefore, in this study we measured SOC density (C amount per unit area) of eight
permanent forest plots from tropical, subtropical, temperate, and boreal forests in China
during two periods in the 1990s and 2010s to quantify their SOC changes. We then analyzed
the potential biotic and climatic drivers in the SOC dynamics across these forests. Finally, we
assessed the changes in SOC stocks in China's forests using the site data obtained from this
study.

**2    Materials and methods**
**2.1    Study sites**
We investigated eight permanent forest plots in four forest sites (from north to south: Great
Xing'anling, Mt. Dongling, Mt. Dinghu, and Jianfengling) (Fig. 1). The four sites spanned a
wide range from 18.7 °N to 52.6 °N in latitude, and belonged to boreal, temperate, subtropical,
and tropical climate zones, respectively, with a climatic difference of approximately 26 °C in
mean annual temperature and 1,200 mm in mean annual precipitation. The eight plots
comprised a boreal larch forest (*Larix gmelinii*), two temperate deciduous broadleaved forests
(*Betula platyphylla* and *Quercus wutaishanica*), a temperate pine plantation (*Pinus*
*tabuliformis*), a subtropical evergreen broadleaved forest, a subtropical pine plantation (*P.*
*massoniana*), a subtropical pine and broadleaved mixed forest, and a tropical mountain
rainforest (for details, see Table 1).
Stand characteristics of all eight plots are summarized in Table 1. The boreal larch forest
was a 100-year-old mature stand at the time of the first sampling (Wang et al., 2001). Three
temperate forest plots (birch, oak, and pine forests) were located along an elevation gradient
on Mt. Dongling, Beijing. Both birch and oak forest plots were 55-year-old secondary forests
at the time of the first sampling, dominated by *B. platyphylla* and *Q. wutaishanica*,

respectively. The temperate pine plantation was 30 years old at the time of the first sampling, and was dominated by *P. tabuliformis* (Fang et al., 2007). Three subtropical forest plots were located in Dinghu Biosphere Reserve in Guangdong Province, South China (Zhou et al., 2006). The subtropical evergreen broadleaved forest was an old-growth stand more than 400 years old, co-dominated by *Castanopsis chinensis*, *Canarium pimela*, *Schima superba*, and *Engelhardtia roxburghiana*. The subtropical pine (*P. massoniana*) plantation was approximately 40 years old at the time of the first sampling. The mature mixed pine and broadleaved forest was approximately 110 years old at the time of the first sampling, and represented the mid-successional stages of monsoon evergreen broadleaved forest in this region. The tropical mountain rainforest plot was located at the Jianfengling National Natural Reserve, southwestern Hainan (Zhou et al., 2013). It had not been disturbed for more than 300 years, and was dominated by species in the families Lauraceae and Fagaceae, such as *Mallotus hookerianus*, *Gironniera subaequalis*, *Cryptocarya chinensis*, *Cyclobalanopsis patelliformis* and *Nephelium topengii*. For detailed descriptions on these eight plots, see Supplementary Materials and Methods.

**2.2 Soil sampling and calculation of SOC content**

The first sampling was conducted between 1987 and 1998 in each of the eight forests (Table 1). We re-measured the same sample plots in each forest between 2008 and 2014 using identical sampling protocols.

In each forest plot, 2–5 pits were dug to collect soil samples for analyzing the physical and chemical properties during the two sampling periods (most in the 1990s during the first sampling period and in the 2010s during the second sampling period). The samples were taken at depth intervals of 10 cm down to the maximum soil depth. In brief, for the boreal forest, three soil pits were established down to the 40-cm soil depth in random locations in the

growing season in 1998. In August 2014, three soil pits were again randomly excavated to the
same soil depth to allow sampling for SOC content and bulk density. For the three temperate
forests, two soil profiles (100 cm depth) were dug in each plot to collect soil samples at 10 cm
intervals during the summer of 1992. In the summer of 2012, three soil profiles were dug, and
soils were sampled from the same horizons in each soil profile (Zhu et al., 2015). The first
sampling in the three subtropical forests was conducted in September 1988 in the evergreen
and pine plots, and in 1987 for the mixed plot, both at the end of the rainy season and at the
beginning of the dry season. Five soil pits (60 cm depth) were randomly excavated to collect
samples for the calculation of SOC content and bulk density. In September 2008, the soil
sampling was repeated. For the tropical forest, five soil profiles (100 cm depth) were
established at 10 cm intervals during summer 1992 and again in summer 2012.
We used consistent sampling and analysis approaches to determine the bulk density and
SOC content between the two sampling times. Three bulk density samples were obtained for
each layer using a standard container 100 cm$^3$ in volume. The soil moisture was determined
by weighing to the nearest 0.1 g after 48 h oven-drying at 105 ℃. The bulk density was
calculated as the ratio of the oven-dried mass to the container volume. Another three paired
samples for C analysis were air-dried, the fine roots removed by hand, and sieved (2 mm
mesh). The SOC content was measured using the wet oxidation method (Nelson and Sommers,
1982) and was calculated according to Eq. (1):
$$\mathrm{SOC} = \sum_{i=1}^{n} CC_i \times Bd_i \times V_i \times HF_i \tag{1}$$
where $CC_i$, $Bd_i$, and $V_i$ are SOC content (%), bulk density (kg m$^{-3}$), and volume (m$^3$) at the
$i$-th soil horizon, respectively. $HF_i$ is calculated as $1 - \dfrac{\text{stone volume+root volume}}{V_i}$ and is a
dimensionless factor that represents the fine soil fraction within a certain soil volume.

**2.3   Calculation of above-ground biomass (AGB) and net primary production**
Diameter at breast height (DBH, 1.3 m) and height of all living trees with DBH > 5 cm were
measured in each plot in the 1990s and 2010s. The AGB of different components (stem, bark,
branches, and foliage) was estimated for all tree species using allometric equations (Table S1).
A standard factor of 0.5 was used to convert biomass to C (Leith and Whittaker, 1975). The
net increment of AGB ($_\Delta$Store) was calculated for each plot as the difference between the
biomass in the 1990s and the 2010s. The above-ground net primary production (ANPP, kg C
ha$^{-1}$ yr$^{-1}$) was calculated from Eq. (2):
$$ANPP = Litterfall + \Delta Store + Mortality \qquad (2)$$
where Litterfall and $\Delta$Store are litter production and above-ground net biomass increment per
year, respectively. Mortality (defined as above-ground dead wood production) was estimated
as the summed production of fallen logs and standing snags per year.

**2.4   Litter and fallen log production**
Annual litterfall was collected from June 2010 to June 2013 in the tropical sites; from June
1990 to June 2008 in the subtropical sites; from April to November 2011–2014 in the
temperate sites; and from May to October 2010–2014 in the boreal sites. Litter (leaves,
flowers, fruits, and woody material < 2 cm diameter) was collected monthly from 10–15 litter
traps (1 $\times$ 1 m$^2$, 1 m above ground) in each plot to calculate annual litter production. After
collection, the samples were taken to the laboratory, oven-dried at 65 ℃ to a constant mass
and weighed. The 10–15 replicates from each plot were averaged as the monthly mean value.
Annual litter production (kg C ha$^{-1}$ yr$^{-1}$) was estimated as the sum of the monthly production
in the year of collection.
Log production represents the mortality (that is, death of entire trees) per year. Annual
log production was determined from 2010 to 2013 in tropical sites; from 1989 to 1996 in
subtropical sites; from 2011 to 2014 in temperate sites; and from 2010 to 2014 in boreal sites.
Stocks of fallen logs were harvested and weighed during each investigated year.

**2.5   Forest area and fossil fuel emission data**
To calculate the amount of C sequestration in China's forest soils, we estimated the changes in
the national forest SOC stocks. We used the mean SOC accumulation rates obtained from this
study and the data of forest area for each forest type documented in the national forest
inventory in 1989–1993, which approximates the first sampling period in the present study
(Guo et al., 2013). The changes in national forest SOC stock were calculated as the product of
SOC density, SOC density change rate, and forest area for major forest types during the
period 1989–1993. In addition, to evaluate the relative importance of forest soil C
sequestration in the national C budget, we obtained the data of fossil fuel emissions during
1991–2010 from the Carbon Dioxide Information Analysis Center (Zheng et al., 2016).

**3   Results**
**3.1   Changes in SOC**
SOC stocks were investigated in eight permanent forest plots in four forest sites from northern
to southern China, in two periods: the 1990s and 2010s. The changes in SOC contents, bulk
density, and SOC stocks in the top 20 cm soil layer between the 1990s and the 2010s are
shown in Fig. 2, Fig. S1 and Fig. S2. The paired *t*-test analysis indicated that SOC contents in
the 0–20 cm depth was significantly higher in the 2010s than in the 1990s (3.2±0.7% vs.
2.9±0.6%; $t = -5.65$, $P < 0.001$) (Table 2). The average rate of increase in SOC content was
0.02% $yr^{-1}$ in the top 20 cm depth, ranging from 0.01% $yr^{-1}$ to 0.04% $yr^{-1}$ across the study
sites. These rates of increase in SOC content in the 0–10 cm horizon (0.03±0.02% $yr^{-1}$) were
three times larger than those in the 10–20 cm horizon (0.01±0.01% $yr^{-1}$) (Table S2). At the
same time, the bulk density of the top 20 cm soil layer decreased in most sites (6 of 8 sites),
with an average rate of decrease of 2.7±3.7 mg cm$^{-3}$ yr$^{-1}$ (Table S3). As a result, the SOC
stock in the top 20 cm soil layer was found to have increased significantly in the past two
decades ($t$ = -5.85, $P$ < 0.001, Table 2), with an average accumulation rate of 332.4±200.2 kg
C ha$^{-1}$ yr$^{-1}$ (0.7±0.4% yr$^{-1}$; Fig. 2; also see Table S3). The temperate pine plantation
experienced the largest increase in SOC stock in the top 20 cm depth (630.8±111.2 kg C ha$^{-1}$
yr$^{-1}$). In contrast, the smallest rate of increase was observed in the subtropical mixed forest
(117.3±25.2 kg C ha$^{-1}$ yr$^{-1}$). It should be noted that SOC stock in the top 20 cm depth in the
subtropical evergreen old-growth forest increased from 35.6±6.0 Mg C ha$^{-1}$ in 1988 to
45.6±6.9 Mg C ha$^{-1}$ in 2008 (increased by 498.3±78.8 kg C ha$^{-1}$ yr$^{-1}$), which led to the highest
relative accumulation rate (1.4±0.2% yr$^{-1}$) among the study sites.

We further compared SOC stocks of the whole soil profile between 1990s and 2010s at a

depth of 0–40 cm in the boreal site, 0–60 cm in the subtropical site, and 0–100 cm in the
temperate and tropical sites (Fig. 3). The SOC stocks of all sampling sites in the 2010s were
higher than those in the 1990s. The paired $t$-test analysis revealed a significant increase in
SOC stocks for the whole soil profile during the sampling period ($t$ = -4.15, $P$ < 0.01; Table 2).
The mean SOC stocks of the whole soil profile in the eight forests increased from 125.2±85.2
Mg C ha$^{-1}$ in the 1990s to 133.6±83.1 Mg C ha$^{-1}$ in the 2010s, with an accumulation rate of
421.2±274.4 kg C ha$^{-1}$ yr$^{-1}$ and a relative increase rate of 0.6±0.5% (Fig. 2). The SOC
accumulation rates displayed large variability among different climate zones and forest types.
For different climate zones, the SOC accumulation rates in the subtropical and tropical sites
were relatively higher than those in the boreal and temperate sites (Fig. 3). The greatest
increase in SOC stock occurred in the subtropical evergreen old-growth forest (907.5±60.1 kg
C ha$^{-1}$ yr$^{-1}$), and the least in the temperate deciduous oak forest (127.2±25.3 kg C ha$^{-1}$ yr$^{-1}$;
Table S3). The relative rates of increase in the subtropical evergreen old-growth forest
(1.3±0.1% yr$^{-1}$) and the subtropical mixed forest (1.5±0.2% yr$^{-1}$) were higher than those in the
temperate forests ($0.1\pm0.0\%$ yr$^{-1}$ in the oak forest, $0.1\pm0.0\%$ yr$^{-1}$ in the pine forest, and
$0.2\pm0.0\%$ yr$^{-1}$ in the birch forest; Table S3).
In addition, the rates of SOC increase ($127.2$–$907.5$ kg C ha$^{-1}$ yr$^{-1}$) was equivalent to $3.6$–
$16.3\%$ of ANPP ($3340.1$–$6944.7$ kg C ha$^{-1}$ yr$^{-1}$), with the highest rate in the subtropical
evergreen forest ($16.3\pm4.2\%$) and the lowest in the temperate oak forest ($3.6\pm3.4\%$) (Tables 3
and S4).

**3.2   Relationships between SOC change rates and biotic and climatic variables**
To understand the possible mechanisms for the rates of SOC increase as described above, we
analyzed the driving forces for this significantly increased SOC stock using measurements of
AGB growth rate, above-ground litter and fallen log production, and ANPP (Table 3). The
linear regression analysis showed that there was no significant correlation between SOC
change rates and AGB growth rate ($P > 0.05$; Fig. 4a). The SOC accumulation rates were
positively and significantly associated with annual litter ($R^2 = 0.66$, $P = 0.01$; Fig. 4b) and
fallen log production ($R^2 = 0.69$, $P = 0.01$; Fig. 4c). The SOC accumulation rates across these
forests were closely associated with the observed ANPP ($R^2 = 0.55$, $P = 0.03$; Fig. 4d), and
also showed an increasing trend with increasing mean annual temperature and precipitation,
despite insignificant (both $P > 0.1$; Figs. 4e and 4f). The multiple regression analysis indicated
the relative effects of biotic factors (AGB growth rate, litter and fallen log production) and
climatic factors (mean annual temperature and precipitation) on the rates of SOC increase (Fig.
4g). When the effects of climatic factors were under control, the biotic factors independently
explained $56.4\%$ of the variations. By comparison, when the effects of biotic factors were
under control, only $7.5\%$ of the variations were explained by the climatic factors.

**4   Discussion**

## 4.1 SOC accumulation

Previous evidence of forest SOC changes comes mainly from individual experiments (Prietzel et al., 2006; Kiser et al., 2009; Häkkinen et al., 2011) or regional comparisons (Lettens et al., 2005; Pan et al., 2011; Ortiz et al., 2013) in European and American forests. In this study, we performed a broad-scale forest soil resampling to evaluate changes in SOC stock across eight permanent forest plots in China. Our measurements suggest that SOC stocks exhibited a significant accumulation in these forests from the 1990s to the 2010s, at the accumulation rate of 127.2–907.5 kg C ha$^{-1}$ yr$^{-1}$. These accumulation rates are comparable to those of other studies that were primarily conducted in boreal and temperate forests in other regions (-11.0–812.0 kg C ha$^{-1}$ yr$^{-1}$, Fig. 5). In detail, the rate of SOC accumulation of the boreal forest in the present study was estimated as 243.4 kg C ha$^{-1}$ yr$^{-1}$, which was within the range of boreal forests in European and American forests (115.6–740.0 kg C ha$^{-1}$ yr$^{-1}$) (Prietzel et al., 2006; Häkkinen et al., 2011; Rantakari et al., 2012; Chapman et al., 2013; Schrumpf et al., 2014). The rates of SOC accumulation in the three temperate forests ranged from 127.2 to 390.8 kg C ha$^{-1}$ yr$^{-1}$, comparable to the regional comparison data of 200.0 kg C ha$^{-1}$ yr$^{-1}$ in the temperate forests of China (Yang et al., 2014). Evidence from soil inventory-based studies of SOC dynamics also demonstrated that soil of boreal and temperate forests in European countries is likely to accumulate C (Berg et al., 2009; Nielsen et al., 2012; Grüneberg et al., 2014). The mean rate of SOC accumulation in the humus layers of boreal forests in Sweden was estimated to be 251.0 kg C ha$^{-1}$ yr$^{-1}$ during the period 1961–2002 (Berg et al., 2009). Nielsen et al. (2012) assessed the rates of SOC change in Denmark's broadleaved deciduous and coniferous forests by two soil inventories conducted during 1990 and 2005. The estimated rates of SOC change in the broadleaved and coniferous forests were 90.0 and 310.0 kg C ha$^{-1}$ yr$^{-1}$, respectively. Two soil inventories provided data for analysis of the mineral soils of forests in Germany, which were found to have sequestrated 410.0 kg C ha$^{-1}$ yr$^{-1}$ during the

period of 1987–2008 (Gr üneberg et al., 2014). Therefore, evidence from long-term
observations, and from the repeated soil sampling in individual studies and in national soil
inventory reports, suggests that soils of boreal and temperate forests in the northern
hemisphere have functioned as C sinks during past decades.

In other subtropical and tropical forest ecosystems, direct evidence of SOC dynamics is

relatively scarce. However, based on the estimates from regional comparisons, Pan et al.
(2011) showed that global tropical forests were a source of 1.4 Pg C ha$^{-1}$ yr$^{-1}$ from 1990 to
2007. At the global scale, tropical land-use changes have caused a sharp drop in forest area,
which also led to a large release of C from tropical forest soils. Without land-use change and
deforestation, soils in subtropical and tropical forests have been functioning as a considerable
C sink during the past two decades in this study (627.6±370.1 and 397.9±84.2 kg C ha$^{-1}$ yr$^{-1}$,
respectively, Table 3). Limited forest management (e.g., litter and dead wood harvest), as well
as catastrophic land-use changes, can result in the loss of C from forest soil. Prietzel et al.
(2016) reported a large loss of SOC in forests in the German Alps, where half of the woody
biomass and dead wood had been harvested over recent decades. On the one hand, harvesting
the forest floor can decrease litter and dead wood inputs into soils and subsequently lead to
the loss of soil C (Davidson and Janssens, 2006). On the other hand, a decrease in the amount
of the forest floor may lead to an increase in soil erosion, especially in mountain forests
(Evans et al., 2013). Additionally, high-elevation ecosystems are expected to be more
sensitive to warming than other regions, with associated changes in soil freezing and thawing
events and in snow cover, which may be another reason for the SOC losses in forests in the
German Alps.

**4.2   Links between biotic and climatic factors and in SOC accumulation**
The forest biomass of China has functioned as a significant C sink over recent decades (Pan et
al., 2011; Fang et al., 2014, 2018). The increase in C accumulation by vegetation supplied
more C inputs into soils, including inputs of litter, woody debris, and root exudates, and
resulted in SOC accumulation (Zhu et al., 2017). However, the rate of SOC change did not
increase with the rate of biomass change in this study (Table S4). We found that soil in the
subtropical old-growth forest increased at the highest sink rate of $907.5 \pm 60.1$ kg C ha$^{-1}$ yr$^{-1}$,
but that vegetation functioned as a significant C source ($-1000.3 \pm 78.2$ kg C ha$^{-1}$ yr$^{-1}$). This
was because the relatively higher annual litterfall and fallen log production occurred in the
old-growth forest, which subsequently resulted in soil C accumulation (Fig. 4). The positive
(but not significant) trend between climatic factors and SOC dynamics may largely be
induced by the internal correlations between climatic and biotic factors (Fig. 4).
The heterotrophic respiration of global forest soil has increased significantly over past
decades (Bond-Lamberty et al., 2018), suggesting that the increment in the rate of soil C input
outweighs that of the rate of soil C output. The increasing heterotrophic respiration of forest
soil is mainly due to ongoing climate change, and especially to increasing temperature. The
increment in forest growth rate is due to increasing temperature, together with increasing $CO_2$
and nitrogen fertilization (Norby et al., 2010; Feng et al., 2019). Thus, the sensitivity of forest
net primary production to ongoing climate change should outweigh that of respiration. We
also found that SOC stock increased from 68.4 Mg C ha$^{-1}$ to 86.6 Mg C ha$^{-1}$, albeit the
biomass C stock decreased significantly from 1988 to 2008 in the subtropical old-growth plot.
The greatest amount of litter and dead wood production and standing crop occurred in the
old-growth plot, which resulted in relatively higher soil C sequestration in the old-growth plot
compared to other plots (Fig. 4, Table S4). Biotic factors explained the variation in SOC
dynamics better than climatic factors. In this study, we did not, however, measure
root-derived C inputs to SOC, although below-ground production also makes a significant
contribution to SOC accumulation (Nadelhoffer and Raich, 1992; Majdi, 2001; Pausch and

Kuzyakov, 2018). Above-ground inputs are mineralized from litter and dead wood, and below-ground inputs may benefit from interactions with soils (Rasse et al., 2005). Even if the effect of climatic factors were controlled and below-ground biotic factors were not included in the analysis, the above-ground biotic factors would explain 56.4% of the variation in the rate of SOC accumulation.

### 4.3 Regional carbon budget

The rate of SOC accumulation ($421.2 \pm 274.4$ kg C ha$^{-1}$ yr$^{-1}$, Fig. 2 and Table S3) is more than one-half of the vegetation C uptake rate in China's forests (702.0 kg C ha$^{-1}$ yr$^{-1}$) (Guo et al., 2013; Fang et al., 2018). This result suggests that China's forest soils have contributed to a negative feedback to climate warming during the past two decades, rather than the positive feedback predicted by coupled C-climate models (Cox et al., 2000; He et al., 2016; Wang et al., 2018).

If we roughly use the inventory-based forest area of 138.8 Mha in China (Guo et al., 2013) and extend the current SOC sink rates obtained in this study to all the forests in the country, China's forest soils have sequestered approximately $1.1 \pm 0.5$ Pg C during the past two decades ($57.1 \pm 26.5$ Tg C yr$^{-1}$). This C accumulation would be equivalent to 2.4–6.8% of the country's fossil $CO_2$ emissions during the contemporary period (1991–2010) (Zheng et al., 2016). By comparing forest SOC data obtained from published literature during the 2000s and a national soil inventory during the 1980s, Yang et al. (2014) estimated significant C accumulation in the forest soils of China. Although they did not estimate the national C budget of these forest soils, we can calculate the national C sequestration rate of forest soil as 67.2 Tg C yr$^{-1}$, based on the C sequestration rates and forest areas of the different forest types in their study. Our results further confirm the assessment, based on repeated measurements at eight permanent forest plots, that soils in China's forests have functioned as a C sink for atmospheric $CO_2$ during the

past two decades.
According to previous estimates, the C sinks of three C sectors: forest vegetation biomass
(Fang et al., 2014), dead wood, and litter (Zhu et al., 2017) during the past two decades were
70.9, 3.9, and 2.8 Tg C yr$^{-1}$, respectively (Table S5). If these previous estimates are
incorporated into the soil C accumulation rate of 57.1$\pm$26.5 Tg C yr$^{-1}$ in the current study, then
China's forests may have sequestered a total of 134.7 Tg C per year between the 1990s and
the 2010s. This is equivalent to 14.5% of the contemporary fossil $CO_2$ emissions in the
country (Zheng et al., 2016). According to the estimate of Pan et al. (2011), the C sink rate of
forests in the temperate regions of the northern hemisphere was 647.1 Tg C yr$^{-1}$. The C
sequestration of China's forests represents 20.8% of the total temperate regions. The
sequestration rate of China's forests is slightly higher than the mean value of the total
temperate regions, relative to the forest area of China (i.e., 18.9% of the forest areas in the
temperate regions). This result indicates that the role of forest soils in the regional C cycle
cannot be ignored, although a large uncertainty about the national C budget of forest soils
remains in our estimates.

**4.4   Uncertainty analysis**
We investigated the SOC stocks in eight permanent plots across four forest biomes in China.
These plots spanned a long-term timescale (approximately 20 years) and a broad spatial scale
(approximately 34 ° of latitude). We also measured several C fluxes (i.e., biomass change rate,
production of litterfall and dead wood) that were relevant to the rate of SOC change. Even so,
the following three factors may introduce uncertainties related to the estimation of SOC
dynamics.
First, the sampling times and intervals between SOC investigations were different across
the sites. The first sampling was performed from 1987 to 1998 and the second was carried out
from 2008 to 2014. As a result, the sampling interval ranged from 16 years in the boreal forest
plot to 21 years in the subtropical mixed forest plot (Table 1). Non-uniform sampling times
and intervals may lead to uncertainties in relation to SOC stocks across the forest plots.

Second, the depth of soil varied substantially, ranging from 40 cm in the boreal site to

100 cm in the temperate and tropical sites. In addition, different numbers (2–5) of soil profiles
were dug in different plots during the first sampling period. To ensure consistency between
the two sampling times, the same number of soil profiles were dug, and in similar locations,
to perform SOC stock investigations during the second sampling period. We performed
continuous observation of litterfall and dead wood production, but the observation times and
durations varied across the plots. Variability in these items may reduce the comparability of
SOC dynamics among plots.

Finally, the rates of SOC change in our study and in inventory-based forest areas and

forest types were used to estimate the C budget of forest soil in China. However, only eight
permanent forest plots were observed in this study, and this will inevitably lead to uncertainty
with respect to national estimations.

**5   Conclusions**
The SOC stocks within the top 20 cm increased by 2.4–12.6 Mg C ha$^{-1}$ across the forests
during the past two decades, with an annual accumulation rate of 332.4±200.2 kg C ha$^{-1}$. If all
soil horizon profiles were included, the soils may have been found to have sequestered 3.6–
16.3% of the annual net primary production across the investigated sites, and the averaged
accumulated rate (421.2 kg C ha$^{-1}$ yr$^{-1}$) may have been more than one-half of the vegetation C
uptake rate (702.0 kg C ha$^{-1}$ yr$^{-1}$) in China's forests. These results demonstrate that these
forest soils have functioned as an important C sink over recent decades, although the
phenomenon may not occur uniformly in forests worldwide. Forest soils store large amounts
of C, and accumulate it steadily and often slowly, but will release it rapidly to the atmosphere
once they are disturbed.

*Data availability.* Allometric equations of above-ground biomass and the data for soil bulk
density, SOC content, stock and their change rates of the eight permanent plots are listed as in
the Supplementary Information. The remaining data that support the findings of this study are
available from the corresponding author upon request.

*Author contributions.* JF designed the research; JZ and JF designed the data analysis. JZ, JF,
ZZ, LJ, XH, HY, GL, CW and GZ performed SOC measurements. JF, YL, CJ and GL
designed sampling and analytical programmes and performed data quality control. JZ, JF, CW,
SZ, PL, JZ, ZT, CZ, RB and YP contributed to the writing of the manuscript.

*Competing interests.* The authors declare no competing interests.

*Financial support.* This work was partly funded by National Key Research and
Development Program of China (2017YFC0503906), National Natural Science Foundation of
China (31700374, 31621091), and the US Forest Service (07-JV-11242300-117).

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

**Table 1.** Location, forest type, mean annual temperature (MAT), and mean annual precipitation (MAP) at eight forest plots in four climate zones, together with forest origin and study periods.

| Site | Forest | Origin | Latitude (°) | Longitude (°) | Elevation (m) | Area (m²) | MAT (°C) | MAP (mm) | Study period |
|------|--------|--------|----------|-----------|-----------|-------------|------|------|--------------|
| Great Xing'anling (Boreal) | Larch | Mature | 52°38'42.06"N | 123°46'7.80"E | 466 | 20×30, 25×40 | -4.3 | 477 | 1998–2014 |
| Mt. Dongling (Temperate) | Birch | Secondary | 39°57'05.82"N | 115°25'38.93"E | 1,350 | 30×35 | 4.7 | 519 | 1992–2012 |
| | Oak | Secondary | 39°57'26.66"N | 115°25'29.14"E | 1,150 | 30×40 | 4.6 | 519 | 1992–2012 |
| | Pine | Plantation | 39°57'33.94"N | 115°25'39.40"E | 1,050 | 20×30 | 5.5 | 506 | 1992–2012 |
| Mt. Dinghu (Subtropical) | Evergreen | Old growth | 23°10'11.21"N | 112°32'21.97"E | 275 | 50×50 | 20.9 | 1698 | 1988–2008 |
| | Mixed | Mature | 23°9'58.51"N | 112°32'23.32"E | 265 | 30×40 | 21.6 | 1680 | 1987–2008 |
| | Pine | Plantation | 23°10'02.75"N | 112°32'30.59"E | 250 | 30×40 | 21.9 | 1677 | 1988–2008 |
| Jianfengling (Tropical) | Evergreen | Old growth | 18°43'47.01"N | 108°53'23.79"E | 870 | 100×100 | 20.6 | 1628 | 1992–2012 |




**Table 2.** Results of the paired-samples *t* tests for soil organic carbon (SOC) content, bulk
density, and SOC stock at different soil depths in the eight forest plots between the 1990s and
the 2010s.

| Soil horizon | SOC content | | | Bulk density | | | SOC stock | | |
|---|---|---|---|---|---|---|---|---|---|
| | *t* | *df* | *P* | *t* | *df* | *P* | *t* | *df* | *P* |
| 0 – 10 cm | -4.22 | 7 | **<0.01** | 2.19 | 7 | 0.06 | -6.50 | 7 | **<0.001** |
| 10 – 20 cm | -4.09 | 7 | **<0.01** | 3.30 | 7 | **<0.05** | -3.26 | 7 | **<0.05** |
| Top 20 cm | -5.65 | 7 | **<0.001** | 1.01 | 7 | **0.35** | -5.85 | 7 | **<0.001** |
| Whole soil profile | - | - | - | - | - | - | -4.15 | 7 | **<0.01** |


**Table 3.** Measured C stocks and fluxes of the four forest sites in China during the 1990s and
the 2010s.

| Parameter | Boreal | Temperate | Subtropical | Tropical |
|---|---|---|---|---|
| **Carbon pool (Mg C ha$^{-1}$)**[*] | | | | |
| AGB | 91.1±25.0 | 89.6±17.4 | 107.0±41.7 | 213.6±41.4 |
| Litter | 4.4±0.0 | 3.9±1.3 | 2.1±0.7 | 1.8±0.2 |
| Dead wood | 1.3±0.5 | 4.5±1.2 | 7.3±6.7 | 5.7±0.8 |
| Soil | 69.4±6.2 | 231.6±14.6 | 67.2±19.5 | 102.6±19.9 |
| Ecosystem total | 166.2±31.7 | 329.6±34.5 | 183.7±68.5 | 323.7±62.3 |
| **Carbon flux (kg C ha$^{-1}$ yr$^{-1}$)** | | | | |
| AGB growth | 899.4±411.0 | 1809.5±521.2 | 798.7±1572.4 | 684.1±145.0 |
| litterfall | 2424.2±283.1 | 1946.7±361.2 | 3385.4±1444.6 | 3970.0±279.8 |
| Fallen log | 13.0±3.7 | 106.1±74.5 | 986.7±967.3 | 1034.2±71.6 |
| Standing snag | 3.5±1.8 | 276.7±111.1 | 220.0±135.7 | 803.4±62.4 |
| ANPP | 3340.1±698.8 | 4139.0±607.7 | 5390.8±1655.3 | 6491.6±559.2 |
| Soil accumulation | 243.4±31.1 | 283.6±138.5 | 627.6±370.1 | 397.9±84.2 |
| Ratio of soil accumulation to ANPP (%) | 7.3±7.8 | 6.7±2.8 (3.6~9.2) | 11.0±5.3 (5.7~16.3) | 6.1±3.3 |

Note: Carbon pool of each ecosystem component at the time of the second sampling (2010s).
AGB, above-ground biomass; ANPP, above-ground net primary production. For details, see
Table S1 in the supplementary information.

 **Figures**

 **Figure 1.** Locations and climatic conditions of the sites. (**a**) Great Xing'anling, the boreal site,

(**b**) Mt. Dongling, the temperate site, (**c**) Mt. Dinghu, the subtropical site, and (**d**) Jianfengling,
the tropical site. The blue and red lines in climatic diagrams are the monthly mean values of
precipitation and temperature, respectively. The blue areas indicate the period in the year
when the precipitation exceeded 100 mm per month. MAT, mean annual temperature; and
MAP, mean annual precipitation.

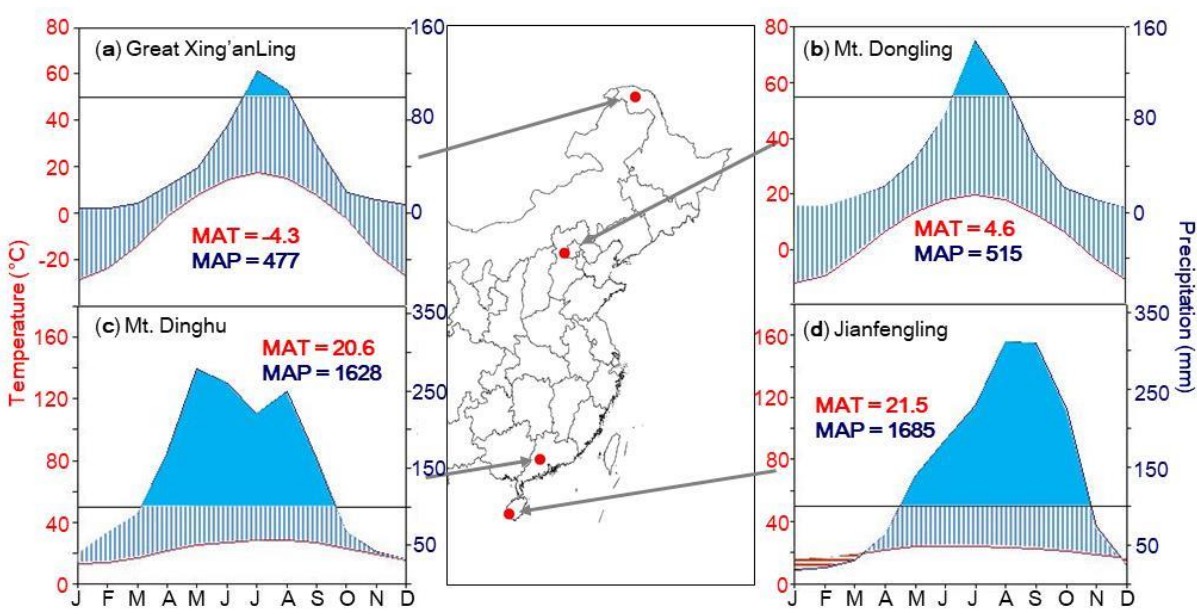



**Figure 2.** Mean soil organic carbon (SOC) content (**a**), bulk density (**b**), SOC stock (**c**) and
their relative change rates (**d**) within 0–20 cm soil depth in the 1990s and the 2010s for the
four forest sites in China. For more details, see Table S2 in the supplementary information.

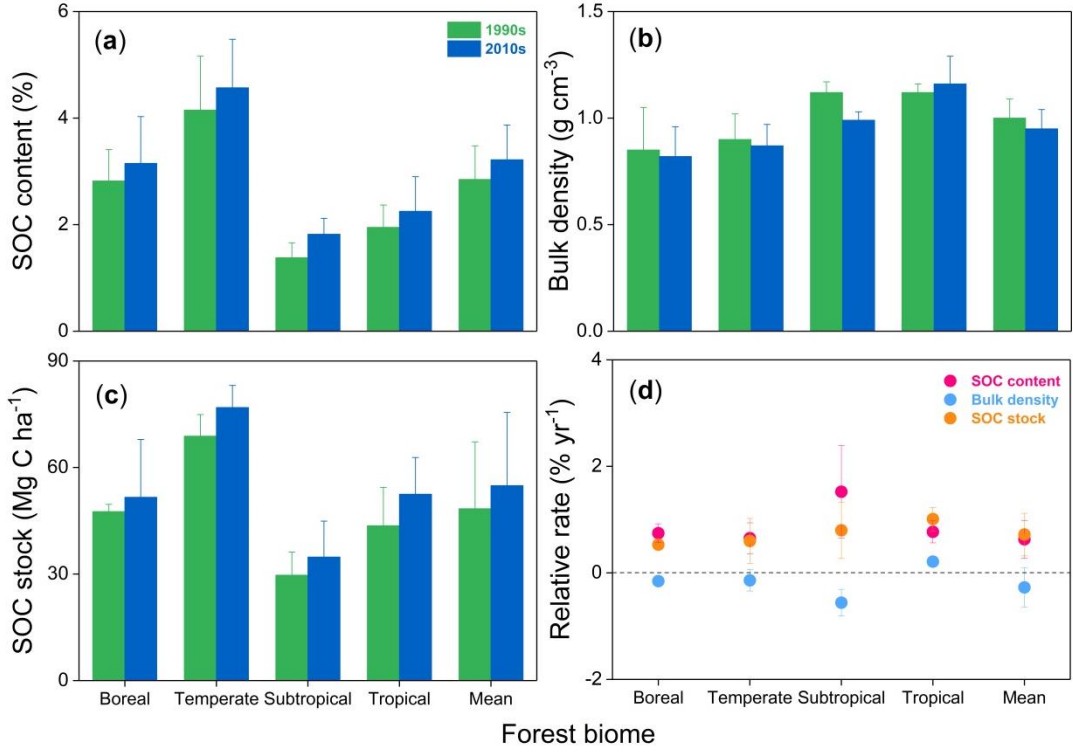



**Figure 3.** Comparison of soil organic carbon (SOC) stocks in eight forest plots in China
between the 1990s and the 2010s. The SOC stocks in all forests during the two periods are
above the 1:1 line, suggesting that all these forests have increased their SOC stock during the
study period. The inset graph shows the SOC sink rates by forest biome (i.e., boreal,
temperate, subtropical, and tropical forests), which are categorized from the eight forest plots.
SOC stocks and change rates are presented as means ±1 SD. For details, see Fig. 1, Table 1,
and Table S1.

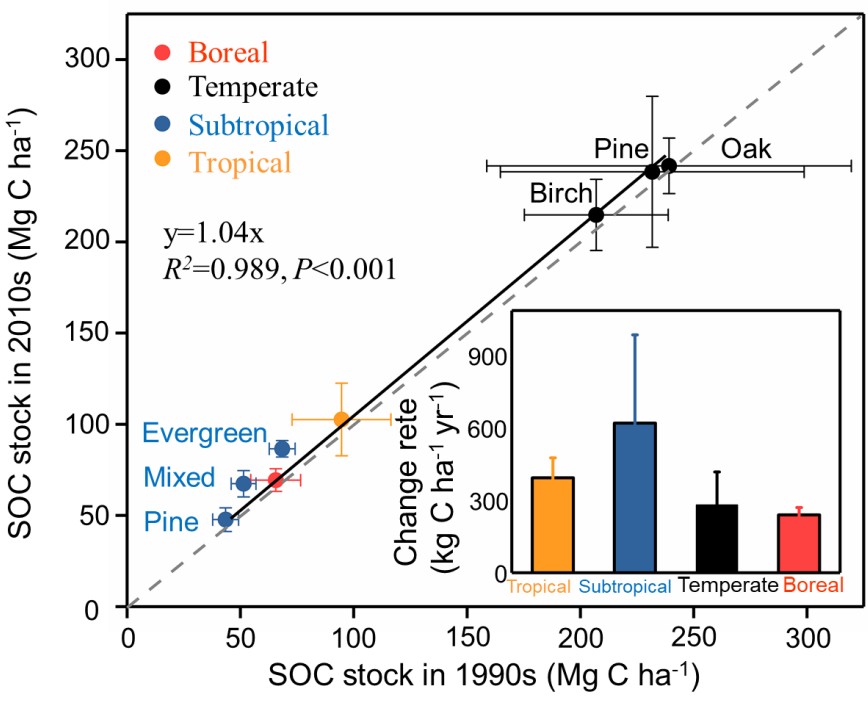



**Figure 4.** Relationships between rates of increase in soil organic carbon (SOC) against biotic and climatic factors in eight forests in China. (**a**)
Biomass increment, (**b**) litter production, (**c**) log production, (**d**) above-ground net primary production (ANPP), (**e**) mean annual temperature
(MAT), (**f**) mean annual precipitation (MAP), and (**g**) the relative effects of biotic (**a**, **b** and **c**) and climatic (**e** and **f**) factors on SOC increase
rates (kg C ha$^{-1}$ yr$^{-1}$) using partial regression analyses. Solid lines indicate significant relationships ($P < 0.05$) and dashed lines represent
insignificant trends ($P > 0.05$) between SOC increase rates and biotic and climatic factors.

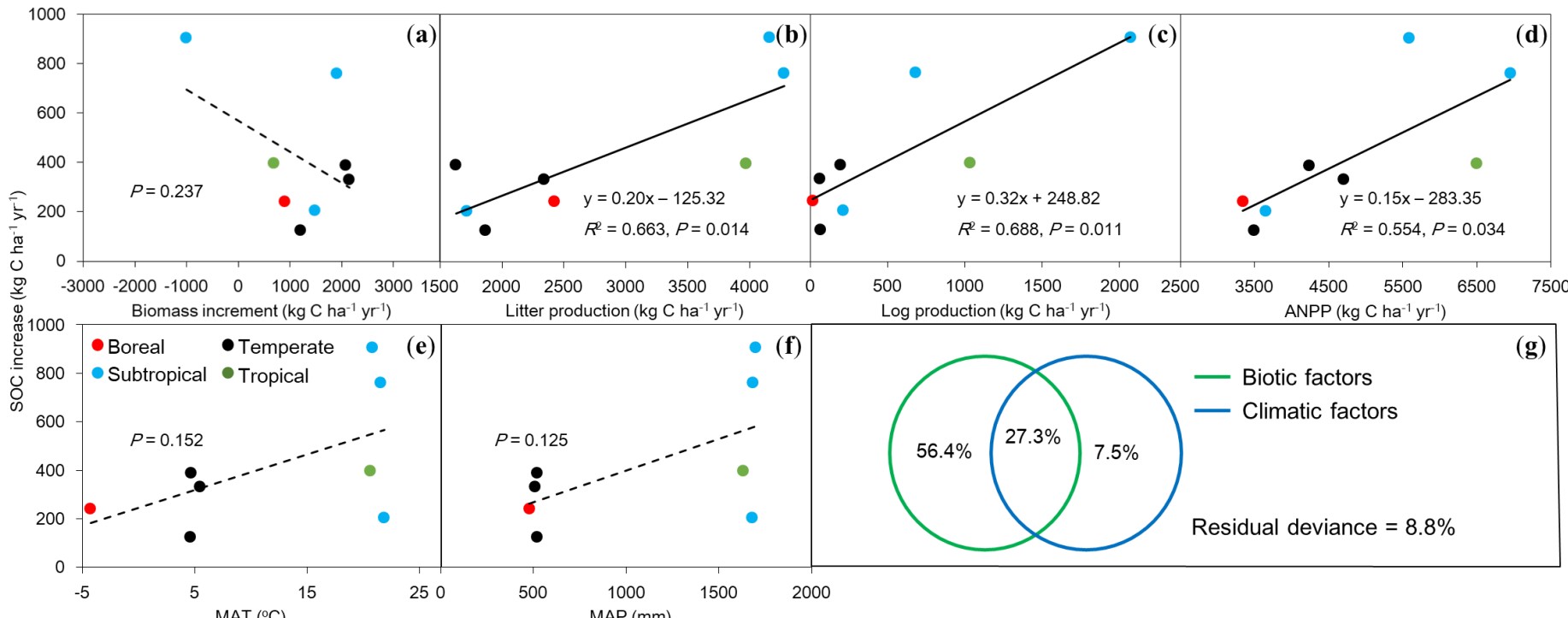



**Figure 5.** Comparison of the changes in forest soil organic carbon (SOC) stocks according to
repeated soil samplings and/or long-term observation. Different colors, shapes, and sizes
represent different forest biomes, ages, and soil depths, respectively. The numbers in
parentheses indicate the sampling times and intervals between the two soil samplings.

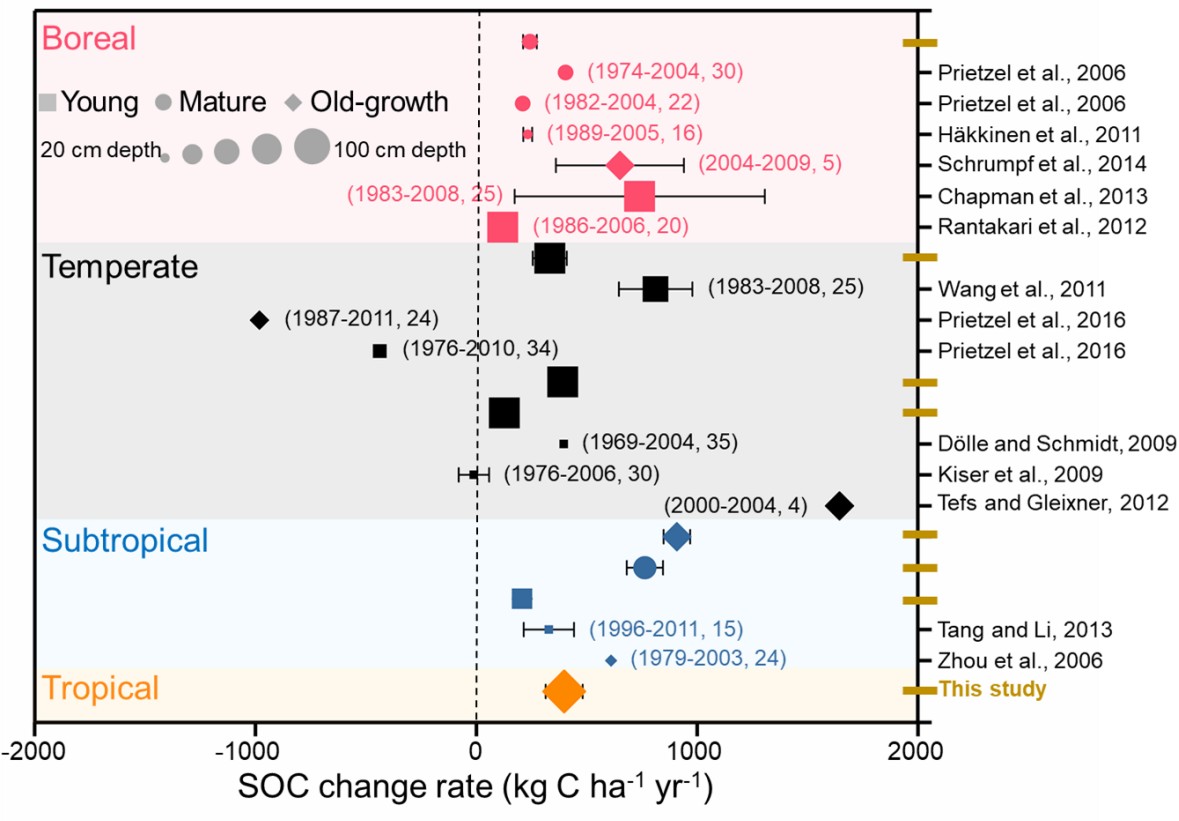
