# Peer review of "Increasing soil carbon stocks in eight permanent forest plots in China"

_Biogeosciences, 2019_

## Referee Comment (RC1) · Anonymous Referee #1 · 20 Sep 2019

Zhu et al. explored an analysis of soil carbon pool in eight permanent plots across China (including primary and secondary forests, and two plantations) in 1990s, and again in 2010s. This resampling enabled them to measure SOC change and change rates, which shows these forest soils were significant carbon sink during the past two decades. The scientific question was quite straightforward, the methods were well established, and the conclusions were reliable and robust. Although the MS is well written, there remain a few minor issues to address (see short list below). but I think these should be straightforward.

L39. Forests have contributed more than half of these carbon (C) fluxes of terrestrial ecosystems. L46. the soil C pool typically has a longer turnover time and higher spatial variability compared to vegetation C pool. L71. SOC density (C stock per unit

area) of eight permanent forest sites. . . L183-184. Use "Four forest sites, eight forest plots" throughout the text. L236. The SOC accumulation rates were positively and significantly associated with annual litterfall and fallen log production. Delete "the above-ground dead organic C production", because only dead plant considered here. L311-333. It is precisely because the data of SOC change is rare. The authors summarized the carbon budget of all components of the forest ecosystem (biomass, soil, litter and dead wood). I suggest that a figure or table should be provided in SI to summarize these results here.

---

## Referee Comment (RC2) · Anonymous Referee #3 · 28 Sep 2019

This manuscript examined SOC dynamics across China's forests, using the direct measurements based on long-term resampling. The scientific question is important and the dataset is unique, and also the manuscript is well written. The following comments should be considered to further improve the manuscript.

Line 71: I think it is better to add one new paragraph to describe the characteristics of China's forests (area, C stock, and the associated environmental change, etc.), and also the related research progress about SOC dynamics across China's forests. The whole logic of the Introduction section will be improved by adding this paragraph.

Line 115: 'in the two sampling periods' should be written as 'during the two sampling periods'. Same issues existed elsewhere.

[Figure]

Line 111-115: From this section, I understand that the sampling interval is largely different among various sites (also see Table 1). The original sampling was conducted during 1987-1998, and the re-sampling was performed during 2008-2014. It is interesting to establish the relationship between the rate of SOC change and sampling interval (or grouping SOC change by sampling interval) to examine its potential effects on SOC dynamics. In addition, I also notice that the sample size within each forest type is different among various sites. Is it possible to examine its potential effects on SOC dynamics?

Line 130-135: Was the same approach also used to determine both the bulk density and SOC content during the original sampling? If so, please clearly describe this point in the revised MS.

Line 156-158/165-167: Again, the sampling period varied substantially among various sites. Please add some descriptions to justify their limited influences on the subsequent data analyses.

Line 171-175: I think the authors need admit the potential uncertainties induced by the limited sample size (8 resampling sites) when upscaling these site-level observations to the national scale. Maybe you can discuss this issue as a potential limitation and also the future directions in the revised MS.

Line 193, and also in Table 2 and Figure 2: It is unclear why the authors focused on 0-20 cm, since 0-30 cm is more popularly used in the literature as the topsoil.

Line 209-211 and also Figure 3: I see that the largest increase was observed in subtropical forests, which had the deepest soils (0-100 cm). Did this pattern also hold true if you compare SOC dynamics within the same soil depth like 0-20 cm? It seems like not, as shown in Figure 2d. Please explained this issue a little bit in the revised MS.

Line 231-246 and also Figure 4: Given that climatic variables did not exert any significant effects on SOC changes (P > 0.05), it might not be appropriate to incorporate

them in the partial regression analysis. Please justify this issue in the revised MS.

Line 254-264 and also Figure 5: Please clearly describe how the authors consider the depth differences when conducting this kind of comparison.

Line 265-275: I am confused about the linkage between these arguments and any results observed in this study. Please clarify.

Line 281 and thereafter: It should be noted that, statistically, the relationships between SOC changes and climatic variables were not significant. To my understanding, it should not put too much efforts to explain those non-significant relationships.

Line 318-321: As mentioned above, uncertainties exist during upscaling. Please discuss this issue in the revised MS.

---

## Referee Comment (RC3) · Anonymous Referee #2 · 3 Oct 2019

Using soil inventory data from four forest sites, authors of this paper explores soil carbon stock change between 1990s and 2010s. They found a significant carbon sink in the forest soils, though magnitude varies greatly. Overall, the manuscript is well written. The core message is clear and contributing to growing knowledge of forest carbon cycling. I believe the manuscript can be accepted for publications after some revision.

The change of soil carbon stock is almost the most uncertain component of ecosystem carbon balance. Although previous studies (e.g. Pan et al., 2011) suggest globally the dominant component of ecosystem carbon sink is in the forest biomass, it is of great interest to compare the sink strength in the soil and in the biomass at different forest ecosystems. Therefore, the authors should compare the strength of the biomass carbon sink and soil carbon sink over these sites, instead of at regional scale with other

inventory data.

Heterotrophic respiration was found to significantly increase at global scale (Bond-lamberty et al., 2018), the existence of soil carbon sink would indicate that the increment of NPP outweighs the increment of HR. This could be further discussed in order to better clarify the processes that contribute to the formation of the carbon sink. The ratio of soil carbon sink to NPP seems very large for some sites, it would be great to extend discussions on why this large ratio of soil sink to NPP is plausible.

Since the soil carbon were measured over four sites (8 plots), it is a bit misleading to call it as "8 typical forests" in the title. The scarcity of available data has made even four sites of data much valuable. There is no need to exaggerate what has been nicely achieved in this study.

In the analyses, it could be interesting to know whether forest types or climatic variations plays a more important role in the size of soil carbon sink. Since some sites only have one plot, it is probably important to further acknowledge this limitations when interpreting the results, which is particularly the case when looking at figure 2.

It is also important to report uncertainties of the magnitude and change of soil carbon stock in figure 3.

The authors spent quite some efforts discussing why their results is in contrast to one study over the Alps. Can the loss of soil carbon in the Alps result from soil erosion? The wood harvests not only reduce the carbon input into the soil, but also expose the soil to erosions, which could be of particular importance in mountainous area. This would be a very interesting discussion since carbon stock change was often treated without considering horizontal soil carbon loss.

The conclusion section also needs some improvements since it highlights the potential role of disturbances, which had not been well discussed or supported before. It is of course reasonable to assume disturbance may affect the soil carbon stock, but the

impacts are very complex and uncertain. The SOC change of protected forests are not very informative to the relationship between disturbances and SOC change, unless further evidences on disturbed forest sites are presented.

———————————————————

---

## Author Comment (AC1) · 13 Nov 2019

**Response to Anonymous Referee #1 on Manuscript bg-2019-319: "*Increasing soil carbon stocks in eight typical forests in China*"**

**General comments**

*Zhu et al. explored an analysis of soil carbon pool in eight permanent plots across China (including primary and secondary forests, and two plantations) in 1990s, and again in 2010s. This resampling enabled them to measure SOC change and change rates, which shows these forest soils were significant carbon sink during the past two decades. The scientific question was quite straightforward, the methods were well established, and the conclusions were reliable and robust. Although the MS is well written, there remain a few minor issues to address (see short list below). but I think these should be straightforward.*

**Response**: Thank you very much for this positive review on our manuscript.

**Specific comments**

*L39. Forests have contributed more than half of these carbon (C) fluxes of terrestrial ecosystems.*

**Response**: This sentence has been modified as you suggested.

*L46. the soil C pool typically has a longer turnover time and higher spatial variability compared to vegetation C pool.*

**Response**: Revised as suggested.

*L71. SOC density (C stock per unit area) of eight permanent forest sites…*

**Response**: We added the missing definition to the text: "Therefore, in this study we measured SOC density (carbon amount per unit area) of eight permanent forest sites from tropical, subtropical, temperate, and boreal forests in China at two periods of the 1990s and 2010s to quantify their SOC changes.".

*L183-184. Use "Four forest sites, eight forest plots" throughout the text.*

**Response**: We checked this throughout the manuscript.

*L236. The SOC accumulation rates were positively and significantly associated with annual litterfall and fallen log production. Delete "the above-ground dead organic C production", because only dead plant considered here.*

**Response**: We agree that only dead wood and plant litter were analyzed in our manuscript. We changed this sentence as you suggested.

*L311-333. It is precisely because the data of SOC change is rare. The authors summarized the carbon budget of all components of the forest ecosystem (biomass, soil, litter and dead wood). I suggest that a figure or table should be provided in SI to summarize these results here.*

**Response**: Thanks for your helpful suggestion, and we will add a new Table into the supplementary information in the new version of the manuscript (Table R1).

Table R1 Summary for C pools and changes in each component of forests in China over the past two decades.

| Component | Carbon pool (Pg C) | Carbon density (Mg ha-1) | National sink (Tg C yr-1) | Source |
|---|---|---|---|---|
| Biomass | 6.87 | 41.3 | 71 | Guo et al. 2013 |
| Soil | 19.98 | 106.1 | 57 | Tang et al., 2018; This study |
| Litter | 0.5 | 3.2 | 3 | Zhu et al. 2017 |
| Dead wood | 0.43 | 2.8 | 4 | Zhu et al. 2017 |
| Ecosystem | 27.35 | 153.4 | 135 | |

---

## Author Comment (AC2) · 13 Nov 2019

**Response to Anonymous Referee #2 on Manuscript bg-2019-319: "*Increasing soil carbon stocks in eight typical forests in China*"**

*Using soil inventory data from four forest sites, authors of this paper explores soil carbon stock change between 1990s and 2010s. They found a significant carbon sink in the forest soils, though magnitude varies greatly. Overall, the manuscript is well written. The core message is clear and contributing to growing knowledge of forest carbon cycling. I believe the manuscript can be accepted for publications after some revision.*

**Response**: Thanks for your positive comments.

*The change of soil carbon stock is almost the most uncertain component of ecosystem carbon balance. Although previous studies (e.g. Pan et al., 2011) suggest globally the dominant component of ecosystem carbon sink is in the forest biomass, it is of great interest to compare the sink strength in the soil and in the biomass at different forest ecosystems. Therefore, the authors should compare the strength of the biomass carbon sink and soil carbon sink over these sites, instead of at regional scale with other inventory data.*

**Response**: We agree that a comparison of biomass and soil carbon sinks across eight forest plots is useful to better classify the carbon budget of the forest ecosystems. Forest biomass of China has functioned as a significant C sink over recent decades (Pan et al., 2011; Fang et al., 2014, 2018). Increased vegetation C accumulation might

produce more C inputs into soils, including inputs of litter, woody debris and root exudates, and then result in SOC accumulation (Schlesinger, 2013; Zhu et al., 2017). However, no statistic relationship between the SOC change rate and biomass change rate was detected across our plots (Fig. R1). We found that SOC stock in the subtropical old-growth forest increased at the highest sink rate of $908\pm60$ kg C ha$^{-1}$ yr$^{-1}$, but the vegetation functioned as a significant C source ($-1000\pm78$ kg C ha$^{-1}$ yr$^{-1}$) during the past two decades. This is probably because the relatively higher mortality created large amount of litterfall and dead wood in the old-growth forest, which subsequently resulted in soil C accumulation (Fig. 4). This result is consistent with a long-term observation in southern China by Zhou et al. (2006) and a global flux synthesis by Luyssaert et al. (2008). We added this discussion into the Section "4.2 Links between biotic and climatic factors and SOC accumulations".

[Figure]

Fig. R1. Relationship between biomass carbon change rate and soil organic carbon

change rate.

*Heterotrophic respiration was found to significantly increase at global scale (Bondlamberty et al., 2018), the existence of soil carbon sink would indicate that the increment of NPP outweighs the increment of HR. This could be further discussed in order to better clarify the processes that contribute to the formation of the carbon sink. The ratio of soil carbon sink to NPP seems very large for some sites, it would be great to extend discussions on why this large ratio of soil sink to NPP is plausible.*

**Response**: We agree that heterotrophic respiration of global forest soil has increased considerably over the recent decades (Bond-Lamberty et al., 2010, 2018). We discussed this and clarified the relationship between SOC sink and NPP, and its potential driving mechanisms in the revised manuscript.

"First, the increasing heterotrophic respiration of forest soil are mainly due to the ongoing climate changes, especially increasing temperature. Whilst the increasing forest biomass are due to the increasing temperature, together with increasing $CO_2$ and nitrogen fertilization (Norby et al., 2010; Feng et al., 2019). Thus, the sensitivity of forest NPP to ongoing climate change should outweigh that of respiration. Second, we found that SOC stock increased from 68.4 Mg C ha$^{-1}$ to 86.6 Mg C ha$^{-1}$, albeit the biomass carbon stock decreased significantly, from 1988 to 2008 in the subtropical old-growth plot. Meanwhile, the highest amount of litter and dead wood production and standing crop occurred in the old-growth plots, which resulted in relatively higher soil carbon sequestration in the old-growth compared to other plots (Figure 4, Table S4)."

*Since the soil carbon were measured over four sites (8 plots), it is a bit misleading to call it as "8 typical forests" in the title. The scarcity of available data has made even four sites of data much valuable. There is no need to exaggerate what has been nicely achieved in this study.*

**Response**: Thanks for this comment, and we changed the title as "Increasing soil carbon stocks in eight permanent forest plots in China" in the updated manuscript.

*In the analyses, it could be interesting to know whether forest types or climatic variations plays a more important role in the size of soil carbon sink. Since some sites only have one plot, it is probably important to further acknowledge this limitations when interpreting the results, which is particularly the case when looking at figure 2.*

**Response**: We agree that the limited observations may induce uncertainties. We added an "Uncertainty analysis" section in the revised manuscript. "The SOC change rates of our study and inventory-based forest area and forest types were used to roughly estimate the carbon budget of forest soil of China's forests. However, only eight permanent forest plots were observed in this study will inevitably lead to uncertainty for national estimate."

*It is also important to report uncertainties of the magnitude and change of soil carbon stock in figure 3.*

**Response**: Thanks for your suggestion. We added this information in Figure 3 in the

revised manuscript (Figure 3).

[Figure]

**Figure 3.** Comparison of soil organic carbon stocks in eight forests of China between the 1990s and the 2010s. The soil organic carbon (SOC) stocks in all forests in the two periods are above the 1:1 line, suggesting that all SOC stocks of these forests have increased during the study period. The inset graph shows the SOC sink rates by forest biomes (i.e., boreal, temperate, subtropical and tropical forests) which are categorized from the eight forests. SOC stocks in this study are presented as means ± 1 SD

*The authors spent quite some efforts discussing why their results is in contrast to one study over the Alps. Can the loss of soil carbon in the Alps result from soil erosion? The wood harvests not only reduce the carbon input into the soil, but also expose the soil to erosions, which could be of particular importance in mountainous area. This would be*

*a very interesting discussion since carbon stock change was often treated without considering horizontal soil carbon loss.*

**Response**: Thanks for your suggestion. Although Prietzel et al. (2016) did not discuss the possibility of soil erosion, we agree that the dead wood harvests might expose the mountain forest soil to erosions. We added the discussion and re-organized this paragraph in the revised manuscript:

"In other subtropical and tropical forest ecosystems, the direct evidence regarding SOC dynamics is relatively scarce. However, based on the estimates from regional comparisons, Pan et al. (2011) showed that tropical forest of the world was a C source of 1.38 Pg C ha$^{-1}$ yr$^{-1}$ from 1990 to 2007. At the global scale, land-use changes have caused a sharp drop in forest area in tropics, which also led to a large C releases in tropical forest soils. Without land-use change and deforestation, soils of the subtropical and tropical forests have functioned as considerable C sink during the past two decades in this study (626±370 and 398±84 kg C ha$^{-1}$ yr$^{-1}$, respectively, Table 3). Not only catastrophic land-use changes, but even slight forest management (e.g. litter and dead wood harvest) can also result in the loss of forest soil carbon. Prietzel et al. (2016) reported a large loss of SOC in forests in the German Alps, where half of the woody biomass and dead wood has been harvested in recent decades. On the one hand, the harvest of forest floor could decrease litter and dead wood inputs into soils and subsequently leads to the loss of soil carbon pool (Davidson and Janssens, 2006). On the other hand, weakened protection of forest floor could lead to increased soil erosion, especially in the mountain forests (Evans et al., 2013). Additionally, the high-elevation

ecosystems are expected to warm more sensitive than other regions with associated changes in soil freezing and thawing events and snow cover, which are probably another reason for the SOC losses of forests in the German Alps."

*The conclusion section also needs some improvements since it highlights the potential role of disturbances, which had not been well discussed or supported before. It is of course reasonable to assume disturbance may affect the soil carbon stock, but the impacts are very complex and uncertain. The SOC change of protected forests are not very informative to the relationship between disturbances and SOC change, unless further evidences on disturbed forest sites are presented.*

**Response**: We agree the comments. We did not analyze the impacts of disturbances on the SOC dynamics in our forest plots. We rephrased the sentence as: "Forest soils store large amounts of C and steadily and often slowly accumulate C, but will rapidly release C to the atmosphere once they are disturbed." in the revised manuscript.

**References**

Bond-Lamberty, B., and Thomson, A.: Temperature-associated increases in the global soil respiration record. Nature, 464, 579–582, https://doi.org/10.1038/nature08930, 2010.

Bond-Lamberty, B., Bailey, V. L., Chen, M., et al.: Globally rising soil heterotrophic respiration over recent decades. Nature, 560, 80–83, https://doi.org/10.1038/s41586-018-0358-x, 2018.

Davidson, E. A., Janssens, I. A.: Temperature sensitivity of soil carbon decomposition and feedbacks to climate change. Nature, 440, 165–173, https://doi.org/10.1038/nature04514, 2006.

Evans, A. M., Perschel, R. T., and Kittler, B. A.: Overview of forest biomass harvesting

guidelines. J. Sustain. Forest., 32, 89–107, https://doi.org/10.1080/10549811.2011.651786, 2013,.

Fang, J., Guo, Z., Hu, H., Kato, T., Muraoka, H., and Son, Y.: Forest biomass carbon sinks in East Asia, with special reference to the relative contributions of forest expansion and forest growth. Glob. Change Biol., 20, 2019–2030, https://doi.org/10.1111/gcb.12512, 2014.

Fang, J., Yu, G., Liu, L., Hu, S., and Chapin III, F. S.: Climate change, human impacts, and carbon sequestration in China. P. Natl. Acad. Sci. USA, 115, 4015–4020, https://doi.org/10.1073/pnas.1700304115, 2018.

Luyssaert, S., Sculze, E. D., Börner, A., Knohl, A., Hessenmöller, D., Law, B. E., Ciais, P., Grace, J.: Old-growth forests as global carbon sinks. Nature, 455, 213–215, https://doi:10.1038/nature07276, 2008.

Norby, R. J., Warren, J. M., Iversen, C. M., Medlyn, B. E., and McMurtrie, R. E.: $CO_2$ enhancement of forest productivity constrained by limited nitrogen availability. P. Natl. Acad. Sci. USA, 107, 19368-19373, 2010.

Pan, Y., Birdsey, R. A., Fang, J. Houghton, R., Kauppi, P. E., Kurz, W. A., Phillips, O. L., Shvidenko, A., Lewis, S. L., Canadell, J. G., Ciais, P., Jackson, R. B., Pacala, S. W., McGuire, A. D., Piao, S., Rautiainen, A., Sitch, S., and Hayes, D.: A large and persistent carbon sink in the world's forests. Science, 333, 988–993, https://doi:10.1126/science.1201609, 2011.

Prietzel, J., Zimmermann, L., Schubert, A., and Christophel, D.: Organic matter losses in German Alps forest soils since the 1970s most likely caused by warming. Nat. Geosci., 9, 543–548, https://doi.org/10.1038/ngeo2732, 2016.

Schlesinger W H, Bernhardt E S. Biogeochemistry: an analysis of global change. Academic press, 2013.

Zhou, G., Liu, S., Li, Z., Zhang, D., Tang, X., Zhou, C., Yan, J., Mo, J.: Old-growth forests can accumulate carbon in soils. Science, 314, 1417, https://doi:10.1126/science.1130168, 2006.

Zhu, J., Hu, H., Tao, S., Chi, X., Li, P., Jiang, L., Ji, C., Zhu, J., Tang, Z., Pan, Y., Birdsey, R. A., He, X., and Fang, J.: Carbon stocks and changes of dead organic matter in China's forests. Nat. Comm., 8, 151, https://doi.org/10.1038/s41467-017-00207-1, 2017.

---

## Author Comment (AC3) · 13 Nov 2019

**Response to Anonymous Referee #3 on Manuscript bg-2019-319: "*Increasing soil carbon stocks in eight typical forests in China*"**

*This manuscript examined SOC dynamics across China's forests, using the direct measurements based on long-term resampling. The scientific question is important and the dataset is unique, and also the manuscript is well written. The following comments should be considered to further improve the manuscript.*

**Response**: Thank you very much for your positive review on our manuscript.

*Line 71: I think it is better to add one new paragraph to describe the characteristics of China's forests (area, C stock, and the associated environmental change, etc.), and also the related research progress about SOC dynamics across China's forests. The whole logic of the Introduction section will be improved by adding this paragraph.*

**Response**: Thank you for your suggestion. We agree that the description for the characteristics of China's forests could improve the logic of the Introduction section. We added a new paragraph in the Introduction section of the revised manuscript:

"Forest in China, with an area of 156 Mha (Guo et al., 2013), span from boreal coniferous forests and deciduous broadleaved forests in the northeast to the tropical rain forests and evergreen broadleaved forests in the south and southwest, covering almost all major forest biomes of the Northern Hemisphere (Fang et al., 2012). Such variations in climate and forest types have provided ideal venues to examine spatial patterns of SOC in relation to meteorological and biological factors. At the national scale, mean

annual air temperature of China has increased by more than 1 °C between 1982 and 2011, which is considerably higher than the global average (Fang et al., 2018). Since the 1980s, the government China has implemented several large-scale National Forest Protection projects. These climatic changes and conservation practices in China have significantly stimulated carbon uptake into forest ecosystem (Fang et al., 2014, 2018; Feng et al., 2019). Several studies have assessed the temporal dynamics in SOC stock across China's forests, using model simulations (Piao et al., 2009; Zhou et al., 2013) or regional assessments (Pan et al., 2011; Yang et al., 2014; Tang et al., 2018). However, these estimates revealed contrasting trends of SOC dynamics and also lacked direct measurements of SOC change."

*Line 115: 'in the two sampling periods' should be written as 'during the two sampling periods'. Same issues existed elsewhere.*

**Response**: Typo corrected throughout the text. Thanks.

*Line 111-115: From this section, I understand that the sampling interval is largely different among various sites (also see Table 1). The original sampling was conducted during 1987-1998, and the re-sampling was performed during 2008-2014. It is interesting to establish the relationship between the rate of SOC change and sampling interval (or grouping SOC change by sampling interval) to examine its potential effects on SOC dynamics. In addition, I also notice that the sample size within each forest type is different among various sites. Is it possible to examine its potential effects on SOC*

*dynamics?*

**Response**: We agree that non-uniform sampling time, interval, size and depth across eight forest plots might lead to possible uncertainties. To examine the possible effects of sampling interval or soil depth, we established the relationship between the sampling interval and soil depth against SOC change rate (Figure R2). However, no significant effects were observed for either sampling interval or the real soil depth on the SOC change rates across plots.

[Figure]

Fig. R2 Effects of sampling interval and real soil depth on the SOC change rates across forest plots.

We added an "Uncertainty analysis" section and discussed the potential influences on the SOC dynamics in the revised manuscript:

"We investigated the SOC stocks at eight permanent plots across four forest biomes in China. These plots spanned a long-term timescale (approximately 20 years) and a broad spatial scale (approximately 34° of latitude). We also measured several carbon fluxes (e.g., biomass change rate, production of litterfall and dead wood) that were relevant to the SOC change rates during the study period. Even so, the following

three aspects may produce uncertainties related to SOC dynamics estimation.

First, the sampling times and interval of SOC investigation were different across the plots. The first sampling was performed during 1987-1998 and the second sampling was carried out during 2008-2014. As a result, the sampling interval ranged from 16 years in boreal forest plot to 21 years in the subtropical mixed forest plot (Table 1). Non-uniform sampling time and interval might lead to uncertainties of SOC stocks across the forest plots.

Second, the real soil depth varied substantially, ranging from 40 cm in the boreal site to 100 cm in the temperate and tropical sites. In addition, different numbers (2-5) of soil profiles for different plots were dug during the first sampling period. To ensure consistency of the two sampling, soil profiles with the same number and similar locations were dug to perform the SOC stocks investigation during the second sampling period. We then performed continuous observation for litterfall and dead wood production, but the observation times and durations varied across the forest plots. Variances of these items might reduce the comparability of SOC dynamics among the plots.

Finally, the SOC change rates of our study and inventory-based forest area and forest types were used to roughly estimate the carbon budget of forest soil of China's forests. However, only eight permanent forest plots were observed in this study will inevitably lead to uncertainty for national estimate."

*Line 130-135: Was the same approach also used to determine both the bulk density and SOC content during the original sampling? If so, please clearly describe this point in*

*the revised MS.*

**Response**: Yes. We used consistent field investigation protocols during the first and second sampling period at the same forest site. We also used consistent sampling and analysis approach to determine soil moisture, organic carbon content and bulk density during two sampling periods. We clarified the description in the revised manuscript.

*Line 156-158/165-167: Again, the sampling period varied substantially among various sites. Please add some descriptions to justify their limited influences on the subsequent data analyses.*

**Response**: we agree that the sampling period of litterfall and tree mortality varied across our forest plots, which could lead to possible uncertainties for the estimate of above-ground net primary production. We also added this discussion into the "Uncertainty analysis" section in the revised manuscript.

*Line 171-175: I think the authors need admit the potential uncertainties induced by the limited sample size (8 resampling sites) when upscaling these site-level observations to the national scale. Maybe you can discuss this issue as a potential limitation and also the future directions in the revised MS.*

**Response**: We admit that the limited number of permanent forest plots may induce uncertainties for the national estimate. We added this discussion into the "Uncertainty analysis" section in the revised manuscript.

*Line 193, and also in Table 2 and Figure 2: It is unclear why the authors focused on 0-20 cm, since 0-30 cm is more popularly used in the literature as the topsoil.*

**Response**: Thank you for this comment. Different studies defined soil at different depths (0-10 cm, 0-20 cm or 0-30 cm) as the surface soil (Fierer et al., 2003; Yang et al., 2014). We used the 0-20 cm as the topsoil because of the following reasons. First, 20 cm soil depth is close to the boundary of the A and B layers across our plots (Wang et al., 2001; Zhou et al., 2006; Zhou et al., 2013; Zhu et al., 2015). Second, we found that the 0-20 cm soil contributed around 80% of carbon sink (332 kg C ha$^{-1}$ yr$^{-1}$) of the whole soil depths (421 kg C ha$^{-1}$ yr$^{-1}$) during the past decades (Table S3).

*Line 209-211 and also Figure 3: I see that the largest increase was observed in subtropical forests, which had the deepest soils (0-100 cm). Did this pattern also hold true if you compare SOC dynamics within the same soil depth like 0-20 cm? It seems like not, as shown in Figure 2d. Please explained this issue a little bit in the revised MS.*

**Response**: We agree that different real soil depth would reduce the comparability of SOC dynamics across our plots. In the revised manuscript, we added corresponding comparation of SOC dynamics within 0-20 cm soil depth as you suggested (Figure R3).

[Figure]

Figure R3 Comparison of soil organic carbon stocks of the surface soil depth (0-20 cm) in eight forests of China between the 1990s and the 2010s. The soil organic carbon (SOC) stocks in all forests in the two periods are above the 1:1 line, suggesting that all these forests have increased their SOC stock during the study period. The inset graph shows the SOC sink rates of the surface soil depth (0-20 cm) by forest biomes.

*Line 231-246 and also Figure 4: Given that climatic variables did not exert any significant effects on SOC changes (P > 0.05), it might not be appropriate to incorporate them in the partial regression analysis. Please justify this issue in the revised MS.*

**Response**: Thanks for this comment. The partial regression analysis showed that only 7.5% of the variations were explained by the climatic factors. This result suggested that climatic factors failed to explain the variances of SOC change rates. The model was only used to compare the relative importance of biotic and climatic factors on SOC

change rate.

*Line 254-264 and also Figure 5: Please clearly describe how the authors consider the depth differences when conducting this kind of comparison.*

**Response**: We agree that different soil depth would reduce the comparability of SOC dynamics. However, measurements of SOC dynamics from permanent forest plots are lacking and inadequate worldwide. The lack of permanent forest sites limited us to compare SOC dynamics at different soil depths and forest types. We clarified the soil depth of all sites in this figure for readability (Figure 5).

*Line 265-275: I am confused about the linkage between these arguments and any results observed in this study. Please clarify.*

**Response**: Sorry for the confusion it caused. In the revised manuscript, we rephrased this paragraph as follows:

"In other subtropical and tropical forest ecosystems, the direct evidence regarding SOC dynamics is relatively scarce. However, based on the estimates from regional comparisons, Pan et al. (2011) showed that tropical forest of the world was a C source of 1.38 Pg C ha$^{-1}$ yr$^{-1}$ from 1990 to 2007. At global scale, land-use changes have caused a sharp drop in forest area in tropics, which also led to a large C releases in tropical forest soils. Without land-use change and deforestation, soils of the subtropical and tropical forests have functioned as considerable C sink during the past two decades in this study (626±370 and 398±84 kg C ha$^{-1}$ yr$^{-1}$, respectively, Table 3). Not only

catastrophic land-use changes, but even slight forest management (e.g. litter and dead wood harvest) can also result in the loss of forest soil carbon. Prietzel et al. (2016) reported a large loss of SOC in forests in the German Alps, where half of the woody biomass and dead wood has been harvested in recent decades. On the one hand, the harvest of forest floor could decrease litter and dead wood inputs into soils and subsequently leads to the loss of soil carbon pool (Davidson and Janssens, 2006). On the other hand, weakened protection of forest floor could lead to increased soil erosion, especially in the mountain forests (Evans et al., 2013). Additionally, the high-elevation ecosystems are expected to warm more sensitive than other regions with associated changes in soil freezing and thawing events and snow cover, which are probably another reason for the SOC losses of forests in the German Alps.".

*Line 281 and thereafter: It should be noted that, statistically, the relationships between SOC changes and climatic variables were not significant. To my understanding, it should not put too much efforts to explain those non-significant relationships.*

**Response**: Thank you for your suggestion. We admit that we discussed too much on the non-significant effects of climatic factors on SOC dynamics. In the revised manuscript, we focused on the influence of biotic factors on the SOC dynamics and reduced the discussions of the relationship between climatic factor and SOC change rate. This paragraph has been re-organized as follows.

"Forest biomass of China has functioned as a significant C sink over recent decades (Pan et al., 2011; Fang et al., 2014, 2018). Increased vegetation C accumulation

produced more C inputs into soils, including inputs of litter, woody debris and root exudates, and resulted in SOC accumulation (Schlesinger, 2013; Zhu et al., 2017). However, the SOC change rate did not increase with the increase of biomass change rate in this study (Table S4). We found that SOC stock in the subtropical old-growth forest increased at the highest sink rate of $908\pm60$ kg C ha$^{-1}$ yr$^{-1}$, but the vegetation functioned as a significant C source ($-1000\pm78$ kg C ha$^{-1}$ yr$^{-1}$). This is because the relatively higher annual litterfall and fallen log production occurred in the old-growth forest, which subsequently resulted in soil C accumulation (Fig. 4). The positive but not significant trend between climatic factors and SOC dynamics could be largely induced by the internal correlations between climatic and biotic factors (Fig. 4)."

*Line 318-321: As mentioned above, uncertainties exist during upscaling. Please discuss this issue in the revised MS.*

**Response**: Thanks for this comment. Uncertainty analysis have been documented comprehensively in the revised manuscript.

**References**

Davidson, E. A., Janssens, I. A.: Temperature sensitivity of soil carbon decomposition and feedbacks to climate change. Nature, 440, 165–173, https://doi.org/10.1038/nature04514, 2006.

Evans, A. M., Perschel, R. T., and Kittler, B. A.: Overview of forest biomass harvesting guidelines. J. Sustain. Forest., 32, 89–107, https://doi.org/10.1080/10549811.2011.651786, 2013,.

Fang, J., Guo, Z., Hu, H., Kato, T., Muraoka, H., and Son, Y.: Forest biomass carbon sinks in East

Asia, with special reference to the relative contributions of forest expansion and forest growth. Glob. Change Biol., 20, 2019–2030, https://doi.org/10.1111/gcb.12512, 2014.

Fang, J., Shen, Z., Tang, Z., et al.: Forest community survey and the structural characteristics of forests in China. Ecography, 35, 1059-1071, https://doi: 10.1111/j.1600-0587.2013.00161.x, 2012.

Fang, J., Yu, G., Liu, L., Hu, S., and Chapin III, F. S.: Climate change, human impacts, and carbon sequestration in China. P. Natl. Acad. Sci. USA, 115, 4015–4020, https://doi.org/10.1073/pnas.1700304115, 2018.

Feng, Y., Zhu, J., Zhao, X., et al.: Changes in the trends of vegetation net primary productivity in China between 1982 and 2015. Environ. Res. Lett., https://doi.org/10.1088/1748-9326/ab4cd8, 2019.

Fierer, N., Bradford, M. A. and Jackson, R. B.: Toward an ecological classification of soil bacteria, Ecology, 88, 1354-1364, https://doi.org/10.1890/05-1839, 2007.

Guo, Z. D., Hu, H. F., Li, P., Li, N. Y., and Fang, J. Y.: Spatio-temporal changes in biomass carbon sinks in China's forests from 1977 to 2008. Sci. China Life Sci., 56, 661–671, https://doi.org/10.1007/s11427-013-4492-2, 2013.

Pan, Y., Birdsey, R. A., Fang, J. Houghton, R., Kauppi, P. E., Kurz, W. A., Phillips, O. L., Shvidenko, A., Lewis, S. L., Canadell, J. G., Ciais, P., Jackson, R. B., Pacala, S. W., McGuire, A. D., Piao, S., Rautiainen, A., Sitch, S., and Hayes, D.: A large and persistent carbon sink in the world's forests. Science, 333, 988–993, https://doi:10.1126/science.1201609, 2011.

Piao, S., Fang, J., Ciais, P., et al.: The carbon balance of terrestrial ecosystems in China. Nature, 458, 1009, https://doi.org/10.1038/nature07944, 2009.

Prietzel, J., Zimmermann, L., Schubert, A., and Christophel, D.: Organic matter losses in German Alps forest soils since the 1970s most likely caused by warming. Nat. Geosci., 9, 543–548, https://doi.org/10.1038/ngeo2732, 2016.

Schlesinger W H, Bernhardt E S. Biogeochemistry: an analysis of global change. Academic press, 2013.

Tang X, Zhao X, Bai Y, et al. Carbon pools in China's terrestrial ecosystems: New estimates based on an intensive field survey. P. Natl. Acad. Sci. USA, 115, 4021-4026,

https://doi.org/10.1073/pnas.1700291115, 2018.

Wang, C., Gower, S. T., Wang, Y., Zhao, H., Yan, P., and Bond-Lamberty, B. P.: The influence of fire on carbon distribution and net primary production of boreal Larix gmelinii forests in north-eastern China. Glob. Change Biol., 7, 719–730, https://doi.org/10.1046/j.1354-1013.2001.00441.x, 2001.

Yang, Y., Li, P., Ding, J., Zhao, X., Ma, W., Ji, C., and Fang, J.: Increased topsoil carbon stock across China's forests. Glob. Change Biol., 20, 2687–2696, https://doi.org/10.1111/gcb.12536, 2014.

Zhou, G., Liu, S., Li, Z., Zhang, D., Tang, X., Zhou, C., Yan, J., Mo, J.: Old-growth forests can accumulate carbon in soils. Science, 314, 1417, https://doi:10.1126/science.1130168, 2006.

Zhou, Z., Jiang, L., Du, E., Hu, H., Li, Y., Chen, D., and Fang, J.: Temperature and substrate availability regulate soil respiration in the tropical mountain rainforests, Hainan Island, China. J. Plant Ecol., 6, 325–334, https://doi.org/10.1093/jpe/rtt034, 2013.

Zhu, J. X., Hu, X. Y., Yao, H., Liu, G. H., Ji, C. J., and Fang, J. Y.: A significant carbon sink in temperate forests in Beijing: based on 20-year field measurements in three stands. Sci. China Life Sci., 58, 1135–1141, https://doi.org/10.1007/s11427-015-4935-z, 2015.

Zhu, J., Hu, H., Tao, S., Chi, X., Li, P., Jiang, L., Ji, C., Zhu, J., Tang, Z., Pan, Y., Birdsey, R. A., He, X., and Fang, J.: Carbon stocks and changes of dead organic matter in China's forests. Nat. Comm., 8, 151, https://doi.org/10.1038/s41467-017-00207-1, 2017.

---

## Author Response (AR1)

November 21, 2019

Dear Dr. Yakov Kuzyakov,

Thank you very much for suggestions and constructive comments from you and the three anonymous reviewers. We have carefully addressed all the comments and suggestions from you and the reviewers, which have been incorporated into the revision of our manuscript.

The point-by-point responses are attached in Response to Referees. We think our revisions are sufficient and thorough, addressing all the questions and issues raised by you and the reviewers. We hope the revised manuscript will satisfy you and the reviewers with all the improvements made. If you have any further concerns and requests, please kindly contact me and we will do our best to fulfil them.

Thank you again for your consideration, and we look forward to hearing from you soon.

Yours sincerely,

Jingyun Fang, Ph.D.

Professor

Department of Ecology

Peking University, Beijing 100871, China

Tel: +86-10-6276 5578, Fax: +86-10-6275 6560

E-mail: jyfang@urban.pku.edu.cn

**Response to Reviewers' Comments on Manuscript bg-2019-319: "*Increasing soil carbon stocks in eight typical forests in China*"**

**Editor Yakov Kuzyakov**

*Further, frequently you use incorrect precision of the values in the Abstract and in the main text, and in Table 3. The precision should correspond to the methods. After improvements, your ms can be accepted.*

**Response**: Thank you very much for your review on our manuscript. Values of SOC stock and change rate are kept in one decimal place throughout the revised manuscript.

**Anonymous Referee #1:**

**General comments**

*Zhu et al. explored an analysis of soil carbon pool in eight permanent plots across China (including primary and secondary forests, and two plantations) in 1990s, and again in 2010s. This resampling enabled them to measure SOC change and change rates, which shows these forest soils were significant carbon sink during the past two decades. The scientific question was quite straightforward, the methods were well established, and the conclusions were reliable and robust. Although the MS is well written, there remain a few minor issues to address (see short list below). but I think these should be straightforward.*

**Response**: Thank you very much for this positive review on our manuscript.

**Specific comments**

*L39. Forests have contributed more than half of these carbon (C) fluxes of terrestrial ecosystems.*

**Response**: This sentence has been modified as you suggested. Please see lines 39-40 in the revised manuscript.

*L46. the soil C pool typically has a longer turnover time and higher spatial variability compared to vegetation C pool.*

**Response**: Revised as suggested (Line 46).

*L71. SOC density (C stock per unit area) of eight permanent forest sites…*

**Response**: We revised this sentence as: "Therefore, in this study we measured SOC density (carbon amount per unit area) of eight permanent forest sites from tropical, subtropical, temperate, and boreal forests in China at two periods of the 1990s and 2010s to quantify their SOC changes." (Lines 86-88).

*L183-184. Use "Four forest sites, eight forest plots" throughout the text.*

**Response**: We checked this throughout the manuscript.

*L236. The SOC accumulation rates were positively and significantly associated with annual litterfall and fallen log production. Delete "the above-ground dead organic C*

*production", because only dead plant considered here.*

**Response**: Done as suggested (Lines 253-254).

*L311-333. It is precisely because the data of SOC change is rare. The authors summarized the carbon budget of all components of the forest ecosystem (biomass, soil, litter and dead wood). I suggest that a figure or table should be provided in SI to summarize these results here.*

**Response**: Thanks for your helpful suggestion. We summarized the national carbon budget in the revised manuscript as you suggested (Table S5).

**Table S5.** Summary for C pools and changes in each component of forests in China over the past two decades.

| Component | Carbon pool (Pg C) | Carbon density (Mg ha$^{-1}$) | National sink (Tg C yr$^{-1}$) | Source |
|---|---|---|---|---|
| Biomass | 6.9 | 41.3 | 71 | Guo et al., 2013 |
| Soil | 20.0 | 106.1 | 57 | Tang et al., 2018; This study |
| Litter | 0.5 | 3.2 | 3 | Zhu et al., 2017 |
| Dead wood | 0.4 | 2.8 | 4 | Zhu et al., 2017 |
| **Ecosystem** | **27.4** | **153.4** | **135** | |

**Anonymous Referee #2**

*Using soil inventory data from four forest sites, authors of this paper explores soil carbon stock change between 1990s and 2010s. They found a significant carbon sink in the forest soils, though magnitude varies greatly. Overall, the manuscript is well written. The core message is clear and contributing to growing knowledge of forest carbon cycling. I believe the manuscript can be accepted for publications after some revision.*

**Response**: Thanks for your positive comments.

*The change of soil carbon stock is almost the most uncertain component of ecosystem carbon balance. Although previous studies (e.g. Pan et al., 2011) suggest globally the dominant component of ecosystem carbon sink is in the forest biomass, it is of great interest to compare the sink strength in the soil and in the biomass at different forest ecosystems. Therefore, the authors should compare the strength of the biomass carbon sink and soil carbon sink over these sites, instead of at regional scale with other inventory data.*

**Response**: Thanks for this suggestion. We discussed the relationship between soil and biomass C sinks across our plots in the section 4.2 *Links between biotic and climatic factors and SOC accumulations.*

Lines 302-312 in the revised manuscript: Forest biomass of China has functioned as a significant C sink over the recent decades (Pan et al., 2011; Fang et al., 2014, 2018).

Increased vegetation-C accumulation supplied more C inputs into soils, including

inputs of litter, woody debris and root exudates, and resulted in SOC accumulation (Zhu et al., 2017). However, the SOC change rate did not increase with the biomass change rate in this study (Table S4). We found that soil in the subtropical old-growth forest increased at the highest sink rate of 907.5 ±60.1 kg C ha$^{-1}$ yr$^{-1}$, but the vegetation functioned as a significant C source (-1000.3 ±78.2 kg C ha$^{-1}$ yr$^{-1}$). This is because the relatively higher annual litterfall and fallen log production occur in the old-growth forest, which subsequently results in soil C accumulation (Fig. 4). The positive but not significant trend between climatic factors and SOC dynamics could be largely induced by the internal correlations between climatic and biotic factors (Fig. 4).

*Heterotrophic respiration was found to significantly increase at global scale (Bondlamberty et al., 2018), the existence of soil carbon sink would indicate that the increment of NPP outweighs the increment of HR. This could be further discussed in order to better clarify the processes that contribute to the formation of the carbon sink. The ratio of soil carbon sink to NPP seems very large for some sites, it would be great to extend discussions on why this large ratio of soil sink to NPP is plausible.*

**Response**: We discussed this and clarified the relationship between SOC sink and NPP, and its potential driving mechanisms in the revised manuscript (Lines 313-324).

The heterotrophic respiration of global forest soil increased significantly over the past decades (Bond-Lamberty et al., 2018), suggesting that the increment of soil carbon input rate outweighs that of soil carbon output rate. The increasing heterotrophic

respiration of forest soil is mainly due to the ongoing climate changes, especially increasing temperature. Whilst the increment of forest growth rate is due to increasing temperature, together with increasing $CO_2$ and nitrogen fertilization (Norby et al., 2010; Feng et al., 2019). Thus, the sensitivity of forest NPP to ongoing climate changes should outweigh that of respiration. Additionally, we found that SOC stock increased from 68.4 Mg C ha$^{-1}$ to 86.6 Mg C ha$^{-1}$, albeit the biomass C stock decreased significantly from 1988 to 2008 in the subtropical old-growth plot. Meanwhile, the highest amount of litter and dead wood production and standing crop occurred in the old-growth plots, which resulted in a relatively higher soil C sequestration in the old-growth plot compared to other plots (Figure 4, Table S4).

*Since the soil carbon were measured over four sites (8 plots), it is a bit misleading to call it as "8 typical forests" in the title. The scarcity of available data has made even four sites of data much valuable. There is no need to exaggerate what has been nicely achieved in this study.*

**Response**: We changed the title as "Increasing soil carbon stocks in eight permanent forest plots in China" in the updated manuscript.

*In the analyses, it could be interesting to know whether forest types or climatic variations plays a more important role in the size of soil carbon sink. Since some sites only have one plot, it is probably important to further acknowledge this limitations when interpreting the results, which is particularly the case when looking at figure 2.*

**Response**: The limited observations may induce uncertainties. We added an "Uncertainty analysis" section in the revised manuscript (Section 4.4, lines 358-380).

*It is also important to report uncertainties of the magnitude and change of soil carbon stock in figure 3.*

**Response**: We updated Figure 3 as you suggested in the revised manuscript.

[Figure]

**Figure 3.** Comparison of soil organic carbon stocks in eight forest plots of China between the 1990s and the 2010s. The soil organic carbon (SOC) stocks in all forests during the two periods are above the 1:1 line, suggesting that all these forests have increased their SOC stock during the study period. The inset graph shows the SOC sink rates by forest biomes (i.e., boreal, temperate, subtropical and tropical forests) which are categorized from the eight forest plots. SOC stocks and change rates are presented as means ± 1 SD. For details, see Figure 1, Table 1 and Supplementary Table 1.

*The authors spent quite some efforts discussing why their results is in contrast to one study over the Alps. Can the loss of soil carbon in the Alps result from soil erosion? The wood harvests not only reduce the carbon input into the soil, but also expose the soil to erosions, which could be of particular importance in mountainous area. This would be a very interesting discussion since carbon stock change was often treated without considering horizontal soil carbon loss.*

**Response**: We added the discussion and re-organized this paragraph in the revised manuscript:

Lines 282-299: In other subtropical and tropical forest ecosystems, the direct evidence regarding SOC dynamics is relatively scarce. However, based on the estimates from regional comparisons, Pan et al. (2011) showed that tropical forest of the world was a C source of 1.38 Pg C ha$^{-1}$ yr$^{-1}$ from 1990 to 2007. At the global scale, tropical land-use changes have caused a sharp drop in forest area, which also led to a large C release in tropical forest soils. Without land-use change and deforestation, soils of the subtropical and tropical forests have been functioning as a considerable C sink during the past two decades in this study (627.6±370.1 and 397.9±84.2 kg C ha$^{-1}$ yr$^{-1}$, respectively, Table 3). Not only catastrophic land-use changes, but also slight forest management (e.g. litter and dead wood harvest) can result in the loss of forest soil carbon. Prietzel et al. (2016) reported a large loss of SOC in German Alps forests, where half of the woody biomass and dead wood have been harvested over the recent decades. On one hand, the harvest of forest floor could decrease litter and dead wood inputs into soils and subsequently lead to the loss of soil carbon (Davidson and Janssens, 2006).

On the other hand, decreased amount of forest floor could lead to an increase of soil erosion, especially in the mountain forests (Evans et al., 2013). Additionally, the high-elevation ecosystems are expected to be more sensitive to warming than other regions with associated changes in soil freezing and thawing events and snow cover, which might be another reason for the SOC losses of German Alps forests.

*The conclusion section also needs some improvements since it highlights the potential role of disturbances, which had not been well discussed or supported before. It is of course reasonable to assume disturbance may affect the soil carbon stock, but the impacts are very complex and uncertain. The SOC change of protected forests are not very informative to the relationship between disturbances and SOC change, unless further evidences on disturbed forest sites are presented.*

**Response**: We rephrased the sentence as: "Forest soils store large amounts of C and accumulate C steadily and often slowly, but will rapidly release C to the atmosphere once they are disturbed." in the revised manuscript (Lines 390-392).

We investigated the SOC stocks at eight permanent plots across four forest biomes

in China. These plots spanned a long-term timescale (approximately 20 years) and a broad spatial scale (approximately 34 ° of latitude). We also measured several C fluxes (i.e., biomass change rate, production of litterfall and dead wood) that were relevant to the SOC change rates. Even so, the following three aspects may produce uncertainties related to the estimation of SOC dynamics.

First, the sampling time and intervals of SOC investigation were different across the sites. The first sampling was performed from 1987 to 1998 and the second sampling was carried out from 2008 to 2014. As a result, the sampling interval ranged from 16 years in boreal forest plot to 21 years in the subtropical mixed forest plot (Table 1). Non-uniform sampling time and interval could lead to uncertainties of SOC stocks across the forest plots.

Second, the depth of soil varied substantially, ranging from 40 cm in the boreal site to 100 cm in the temperate and tropical sites. In addition, different numbers (2-5) of soil profiles for different plots were dug during the first sampling period. To ensure consistency between the two sampling times, soil profiles with the same number and similar locations were dug to perform the SOC stocks investigation during the second sampling period. We performed continuous observation for litterfall and dead wood production, but the observation times and durations varied across the plots. Variances of these items might reduce the comparability of SOC dynamics among plots.

Finally, the SOC change rates of our study and inventory-based forest area and forest types were used to estimate the carbon budget of forest soil of China. However, only eight permanent forest plots were observed in this study will inevitably lead to

uncertainty for national estimations.

*Line 130-135: Was the same approach also used to determine both the bulk density and SOC content during the original sampling? If so, please clearly describe this point in the revised MS.*

**Response**: Yes. We used consistent field investigation protocols during the first and second sampling period at the same forest site. We also used consistent sampling and analysis approach to determine soil moisture, organic carbon content and bulk density during two sampling periods. We clarified the description in the revised manuscript (Lines 146-153).

*Line 156-158/165-167: Again, the sampling period varied substantially among various sites. Please add some descriptions to justify their limited influences on the subsequent data analyses.*

**Response**: The annual fluctuations of leaf and woody litter production exist in this study. We discussed this uncertainty in the section 4.2 Uncertainty analysis in the revised manuscript (Lines 373-376).

*Line 171-175: I think the authors need admit the potential uncertainties induced by the limited sample size (8 resampling sites) when upscaling these site-level observations to the national scale. Maybe you can discuss this issue as a potential limitation and also the future directions in the revised MS.*

**Response**: We admit that the limited number of permanent forest plots may induce

uncertainties for the national estimate. We added this discussion into the section 4.2 Uncertainty analysis in the revised manuscript (Lines 377-380).

*Line 193, and also in Table 2 and Figure 2: It is unclear why the authors focused on 0-20 cm, since 0-30 cm is more popularly used in the literature as the topsoil.*

**Response**: Thank you for this comment. Different studies defined soil at different depths (0-10 cm, 0-20 cm or 0-30 cm) as the surface soil (Fierer et al., 2007; Yang et al., 2014). We used the 0-20 cm as the topsoil because of the following reasons. First, 20 cm soil depth is close to the boundary of the A and B layers across our plots (Wang et al., 2001; Zhou et al., 2006; Zhou et al., 2013; Zhu et al., 2015). Second, we found that the 0-20 cm soil contributed around 80% of carbon sink (332 kg C ha$^{-1}$ yr$^{-1}$) of the whole soil depths (421 kg C ha$^{-1}$ yr$^{-1}$) during the past decades (Table S3).

*Line 209-211 and also Figure 3: I see that the largest increase was observed in subtropical forests, which had the deepest soils (0-100 cm). Did this pattern also hold true if you compare SOC dynamics within the same soil depth like 0-20 cm? It seems like not, as shown in Figure 2d. Please explained this issue a little bit in the revised MS.*

**Response**: We added the comparation of SOC dynamics within 0-20 cm soil depth as you suggested (Figure S2) in the revised Supplementary Information.

[Figure]

**Figure S2**. Comparison of soil organic carbon stocks of the surface soil depth (0-20 cm) in the eight forest plots of China between the 1990s and the 2010s. The inset graph shows the SOC change rates of the surface soil depth (0-20 cm) by forest biomes.

*Line 231-246 and also Figure 4: Given that climatic variables did not exert any significant effects on SOC changes (P > 0.05), it might not be appropriate to incorporate them in the partial regression analysis. Please justify this issue in the revised MS.*

**Response**: Thanks for this comment. The partial regression analysis showed that only 7.5% of the variations were explained by the climatic factors. This result suggested that climatic factors failed to explain the variances of SOC change rates. The model was only used to compare the relative importance of biotic and climatic factors on SOC change rate.

*Line 254-264 and also Figure 5: Please clearly describe how the authors consider the depth differences when conducting this kind of comparison.*

**Response**: Measurements of SOC dynamics from permanent forest plots are lacking and inadequate worldwide. The lack of permanent forest sites limited us to compare SOC dynamics at different soil depths and forest types. In the revised manuscript, we clarified the soil depth of all sites for readability (Figure 5).

[Figure]

**Figure 5**. Comparison of the changes in forest soil organic carbon stocks according to repeated soil samplings and/or long-term observations. Different colors, shapes and sizes represent different forest biomes, ages, and soil depth, respectively.

*Line 265-275: I am confused about the linkage between these arguments and any results*

*observed in this study. Please clarify.*

**Response**: Sorry for the confusion it caused. In the revised manuscript, we rephrased this paragraph in the revised manuscript (Lines 282-299):

In other subtropical and tropical forest ecosystems, the direct evidence regarding SOC dynamics is relatively scarce. However, based on the estimates from regional comparisons, Pan et al. (2011) showed that tropical forest of the world was a C source of 1.38 Pg C ha$^{-1}$ yr$^{-1}$ from 1990 to 2007. At the global scale, tropical land-use changes have caused a sharp drop in forest area, which also led to a large C release in tropical forest soils. Without land-use change and deforestation, soils of the subtropical and tropical forests have been functioning as a considerable C sink during the past two decades in this study (627.6±370.1 and 397.9±84.2 kg C ha$^{-1}$ yr$^{-1}$, respectively, Table 3). Not only catastrophic land-use changes, but also slight forest management (e.g. litter and dead wood harvest) can result in the loss of forest soil carbon. Prietzel et al. (2016) reported a large loss of SOC in German Alps forests, where half of the woody biomass and dead wood have been harvested over the recent decades. On one hand, the harvest of forest floor could decrease litter and dead wood inputs into soils and subsequently lead to the loss of soil carbon (Davidson and Janssens, 2006). On the other hand, decreased amount of forest floor could lead to an increase of soil erosion, especially in the mountain forests (Evans et al., 2013). Additionally, the high-elevation ecosystems are expected to be more sensitive to warming than other regions with associated changes in soil freezing and thawing events and snow cover, which might be another reason for the SOC losses of German Alps forests.

*Line 281 and thereafter: It should be noted that, statistically, the relationships between SOC changes and climatic variables were not significant. To my understanding, it should not put too much efforts to explain those non-significant relationships.*

**Response**: Thank you for your suggestion. In the revised manuscript, we focused on the influence of biotic factors on the SOC dynamics and reduced the discussions of the relationship between climatic factor and SOC change rate. This paragraph has been re-organized (Lines 302-312).

Forest biomass of China has functioned as a significant C sink over the recent decades (Pan et al., 2011; Fang et al., 2014, 2018). Increased vegetation-C accumulation supplied more C inputs into soils, including inputs of litter, woody debris and root exudates, and resulted in SOC accumulation (Zhu et al., 2017). However, the SOC change rate did not increase with the biomass change rate in this study (Table S4). We found that soil in the subtropical old-growth forest increased at the highest sink rate of $907.5 \pm 60.1$ kg C ha$^{-1}$ yr$^{-1}$, but the vegetation functioned as a significant C source ($-1000.3 \pm 78.2$ kg C ha$^{-1}$ yr$^{-1}$). This is because the relatively higher annual litterfall and fallen log production occur in the old-growth forest, which subsequently results in soil C accumulation (Fig. 4). The positive but not significant trend between climatic factors and SOC dynamics could be largely induced by the internal correlations between climatic and biotic factors (Fig. 4).

*Line 318-321: As mentioned above, uncertainties exist during upscaling. Please discuss*

*this issue in the revised MS.*

**Response**: Thanks for this comment. Uncertainty analysis have been documented comprehensively in the revised manuscript (Lines 358-380).

**References**

[revised manuscript text omitted]

Nat. Comm., 8, 151, https://doi.org/10.1038/s41467-017-00207-1, 2017.

---

## Author Response (AR2)

**Responses to review by Editor Yakov Kuzyakov of *Biogeosciences* manuscript bg-2019-319: "*Increasing soil carbon stocks in eight permanent forest plots in China*"**

We are very grateful to editor Dr. Yakov Kuzyakov for the detailed and valuable comments on our manuscript. We think our revisions are sufficient and thorough, addressing all the questions and issues during our revisions.

Editor's comments and our responses are presented below.

Editor's comments are given in *italic font*, and our responses in blue regular font.

*Your submission is not prepared as correct submission:*

*- you have many yellow labelings - surely the first author has not improved the ms according to submission of somebody.*

Response: We must apologize for the yellow labeling in the original manuscript. We are extremely sorry for the misunderstanding this has caused and for not clarifying it in the previous response letters. Actually, we had carefully merged the revisions of the coauthors, and the yellow labels were a way to mark the changes we made in the previous manuscript. For this updated manuscript, we have uploaded a "clean" version (without any track changes) to the system, and have also uploaded a "marked-up" version (showing the changes we made in the revised manuscript) after the "List of all relevant changes made in the manuscript " at the bottom of this file.

*- the precision of the numbers in the text and in the Tables.*

Response: Thank you for this comment. According to our understanding, values of soil organic carbon (SOC) content (%), bulk density (g cm$^{-3}$), SOC stock (Mg C ha$^{-1}$), change rate of bulk density (mg cm$^{-3}$ yr$^{-1}$), change rate of SOC stock (kg C ha$^{-1}$ yr$^{-1}$) and the relative change rate of SOC stock (% yr$^{-1}$) should be given to one decimal place throughout the revised text, in the in-text tables and also in the supplementary tables. However, considering the validity of the data, values of the change rate of SOC content (% yr$^{-1}$) are kept in two decimal place throughout the revised manuscript.

*- the Abstract is very poor.*

Response: We have clarified the methods and results in the section of *Abstract* in the revised manuscript. We added information about the four study sites and the dynamics of SOC content and bulk density during the past two decades.

*- for the Fig 5 - there are much more papers from the literature.*

Response: Thank you for this helpful comment. We have added SOC change rates from four public sources of literature (a boreal site, Rantakari et al., 2012; two temperate sites, Dölle and Schmidt, 2009, and Tefs and Gleixner, 2012; and a subtropical site, Tang and Li, 2013). We have also added information about sampling intervals in each site into Fig. 5 in the revised manuscript.

[Figure]

Fig. 5. Comparison of the changes in forest soil organic carbon (SOC) stocks according to repeated soil samplings and/or long-term observations. Different colors, shapes, and sizes represent different forest biomes, ages, and soil depths, respectively. The numbers in parentheses indicate the sampling times and intervals between the two soil samplings.

*- some of the regressions in Fig 4 are not correct.*

Response: Thank you for this helpful comment. We have deleted the function for the insignificant relationship, and retained the dashed lines to represent insignificant trends, in the new figure. We have clarified this in the figure legend in the revised manuscript.

[Figure]

Figure 4. Relationships between rates of increase in soil organic carbon (SOC) against biotic and climatic factors in eight forests in China. (**a**) Biomass increment, (**b**) litter production, (**c**) log production, (**d**) above-ground net primary production (ANPP), (**e**) mean annual temperature (MAT), (**f**) mean annual precipitation (MAP), and (**g**) the relative effects of biotic (**a**, **b** and **c**) and climatic (**e** and **f**) factors on SOC increase rates (kg C ha$^{-1}$ yr$^{-1}$) using partial regression analyses. Solid lines indicate significant relationships ($P < 0.05$) and dashed lines represent insignificant trends ($P > 0.05$) between SOC increase rates and biotic and climatic factors.

*- if your study is focused only on China, you should not publish this in an international Journal. Make your study internationally interesting.*

Response: Thank you for this important comment. To increase the relevance of our manuscript, especially for international readers, we have made the following improvements in the revised manuscript. (1) We have compared our SOC change rates in boreal and temperate forests with SOC dynamics from national soil inventories of other countries, such as those from Germany, Sweden, and Denmark. (2) We have compared our regional SOC budget estimate with the results of Yang et al. (2014), who estimate SOC dynamics of China's forests by comparing measurements from literature during the 2000s with historical records derived from a national soil inventory during the 1980s. (3) We also compared the regional SOC budget estimate with a study of global forest carbon budgets (Pan et al. 2011). We found that carbon sequestration in China's forests represented 20.8% of the total temperate regions of the northern hemisphere. The sequestration rate in China's forests is slightly higher than the mean value of the total temperate regions, relative to the forest area of China (i.e., 18.9% of the forest areas in the temperate regions). Then we added this information, and its implication for the importance of forest soil sequestration in this region, to the revised manuscript.

[revised manuscript text omitted]